# Current reversal leads to regime change in Amery Ice Shelf cavity in the twenty-first century

Jing Jin[1, a], Antony J. Payne[1, a], and Christopher Y. S. Bull[2, b]

[1]School of Geographical Sciences, University of Bristol, Bristol, UK
[2]Department of Geography and Environmental Sciences, Northumbria University, Newcastle Upon Tyne, UK
[a]now at: Department of Earth, Ocean and Ecological Sciences, University of Liverpool, Liverpool, UK
[b]now at: ACCESS-NRI, Australian National University, Canberra, Australia

**Correspondence:** Jing Jin (Jing.Jin2@liverpool.ac.uk)

**Abstract.** The Amery Ice Shelf (AmIS), the third largest ice shelf in Antarctica, has experienced relatively low rates of basal melt during the past decades. However, it is unclear how AmIS melting will respond to a future warming climate. Here, we use a regional ocean model forced by different climate scenarios to investigate AmIS melting by 2100. The areally-averaged melt rate is projected to increase from 0.7 m·yr$^{-1}$ to 8 m·yr$^{-1}$ in the low-emission scenario or 17 m·yr$^{-1}$ in the high-emission

scenario in 2100. An abrupt increase in melt rate happens in the 2060s in both scenarios. The redistribution of local salinity (hence density) in front of AmIS forms a new geostrophic balance, leading to the reversal of local currents. This transforms AmIS from a cold cavity to a warm cavity, and results in the jump in ice shelf melting. While the projections suggest that AmIS is unlikely to experience instability in the coming century, the high melting draws our attention to the role of oceanic processes in basal mass loss of Antarctic ice shelves in climate change.

## 1  Introduction

The Amery Ice Shelf (AmIS) is a large ice shelf in Antarctica (Figure 1). It is fed by Lambert Glacier system which accounts for about 16% of the grounded ice in East Antarctica (Allison, 1979). AmIS has a deep grounding line reaching ~2500m below sea surface (Galton-Fenzi et al., 2008; Yang et al., 2021; Chen et al., 2023), with an in-situ freezing point ~2 °C lower than the coldest water mass outside of the ice shelf cavity (Galton-Fenzi et al., 2012). This makes AmIS susceptible to

ocean temperature changes in the deep ice shelf cavity (Galton-Fenzi et al., 2012; Wang et al., 2022). As ice shelf melt rates quadratically respond to external thermal forcing (Holland et al., 2008), ice with a low freezing point implies that it is more sensitive to any ocean temperature change than with a higher freezing point.

Prydz Bay has a v-shaped coastline and constrains the AmIS (O'Brien et al., 2014). It is divided by Prydz Channel —a trough with the depth of ~500 m at the shelf break and a deeper trough at the inner embayment named Amery Depression with

bathymetry from 500-1000 m (Figure 1). The Fram Bank and the Four Ladies Bank are located on the western and eastern side of AD, respectively, and exhibit dramatic differences of zonal topography (Figure 1). On the west of Amery Depression, the depth of Fram Bank rises sharply from ~600 m to ~200 m over a distance of ~50 km. On the east of Amery Depression, Four Ladies Bank has a stepwise decrease from the depth of ~200 m to ~500m over a distance of ~200 km (Liu et al., 2018).

High Salinity Shelf Water (HSSW) and modified Circumpolar Deep Water (mCDW) are major water masses causing AmIS basal melting (Galton-Fenzi et al., 2012; Herraiz-Borreguero et al., 2015, 2016). HSSW is a dense and cold (slightly below the surface freezing point) water mass that forms in coastal polynyas within Prydz Bay during sea ice formation (Ohshima et al., 2013; Williams et al., 2016; Portela et al., 2021). When HSSW descends into the AmIS cavity, the temperature of HSSW is higher than the in situ (pressure dependent) freezing point of seawater in the deepest cavity, which results in a basal melt rate of >30 m·yr$^{-1}$ at the grounding line (Galton-Fenzi et al., 2012). The resulting meltwater, named Ice Shelf Water (ISW), upwells along the western ice shelf base. ISW, with a temperature of the in situ freezing point, can be supercooled when it is ascending and refreezing beneath the northwestern AmIS (Craven et al., 2009).

The cavity under AmIS is presently filled with relatively cold HSSW of -2.2°C to -1.8°C (Craven et al., 2004; Herraiz-Borreguero et al., 2015). This results in relatively low basal melting along with basal refreezing beneath the northwestern AmIS (Depoorter et al., 2013; Rignot et al., 2013). Basal melt rates of AmIS have been stable during the past few decades (Adusumilli et al., 2020).

However, Southern Ocean temperatures are projected to increase in a warming climate (Fox-Kemper et al., 2021; Rintoul et al., 2018). The previous studies predict the increased upwelling of mCDW onto the continental shelf of the AmIS sector, resulting in a shift of cavity regime and consequently an increase of basal melting (Naughten et al., 2018; Kusahara et al., 2023; Thomas et al., 2023; Mathiot and Jourdain, 2023). These studies provide insight into the drivers of warming on the continental shelf. For example, a freshening at the surface of Prydz Bay increases vertical stratification and induces warming at depth (Aoki et al., 2022; Thomas et al., 2023; Kusahara et al., 2023). Another mechanism is related to the poleward shift of westerly winds, which enhances the upwelling of mCDW across the shelf break (Spence et al., 2017; Guo et al., 2019; Verfaillie et al., 2022). However, future changes in the links between the warming in PB and the warming in the AmIS ice shelf cavity, in other words, local oceanic currents/intrusive pathways of warm water, still lack investigation.

mCDW was thought to be absent in the AmIS cavity (Craven et al., 2004). However, it has been observed at the AmIS calving front entering the cavity during winter recently (Herraiz-Borreguero et al., 2015). Other hydrographical observations and modelling studies have documented the presence of mCDW on the continental shelf in Prydz Bay (Galton-Fenzi et al., 2012; Herraiz-Borreguero et al., 2015, 2016; Williams et al., 2016; Liu et al., 2017; Guo et al., 2022). The two main pathways of mCDW intruding beneath AmIS are:

1. A large cyclonic gyre encircles Amery Depression, known as Prydz Bay Gyre (PBG) (Smith et al., 1984; Nunes Vaz and Lennon, 1996; Heywood et al., 1999). mCDW upwells across the continental shelf, arriving at Prydz Bay over Four Ladies Bank, and is transported by the eastern branch of PBG toward AmIS (Galton-Fenzi et al., 2012; Herraiz-Borreguero et al., 2015; Williams et al., 2016; Liu et al., 2017). Some ISW exits the cavity and recirculates within PBG and returns beneath AmIS (Williams et al., 2016).

2. A narrow current flows between Four Ladies Bank and the East coast, named Prydz Bay Eastern Coastal Current, which originates in the Antarctic Slope Current (Liu et al., 2017, 2018). Due to the step-like deepened bathymetry over Four Ladies Bank, Antarctic Slope Current is redirected shoreward to conserve potential vorticity, resulting in the formation of Prydz Bay Eastern Coastal Current (Liu et al., 2018), bringing mCDW to Prydz Bay (Liu et al., 2017).

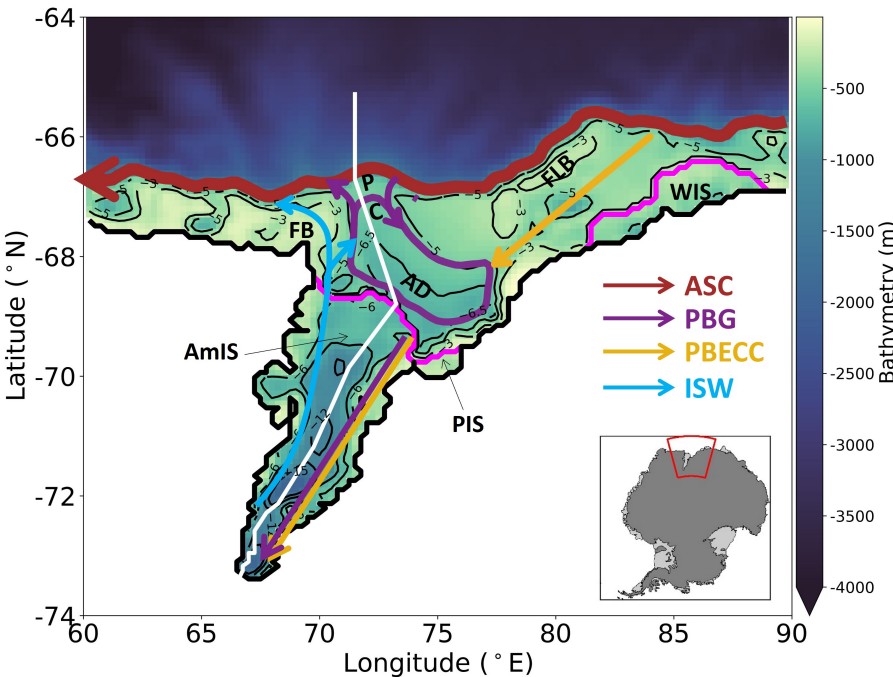

**Figure 1.** Model domain and schematic diagram of ocean circulation in Prydz Bay. The colour scale shows bathymetry (m). The thin black lines on the continental shelf indicate bathymetry of -300, -500 and -650 m. The thin black lines on Amery Ice Shelf (AmIS) show bathymetry of -600, -800, -1200, -1500, -2000 m. The thick black line represents the coastal line, and the thick magenta lines show the ice shelf fronts. The thick red arrow represents Antarctic Slope Current (ASC). The purple arrows show Prydz Bay Gyre (PBG). The yellow arrow indicates Prydz Bay Eastern Coastal Current (PBECC). Thin purple and yellow arrows in AmIS show modified Circumpolar Deep Water. Cyan arrows represent Ice Shelf Water (ISW). The white line shows a transect extending from the grounding line to the shelf break. Geographical features are labelled as follows: AmIS: Amery Ice Shelf, PIS: Publications Ice Shelf WIS: West Ice Shelf, PC: Prydz Bay Channel, AD: Amery Depression, FLB: Four Ladies Bank, FB: Fram Bank. Inset: location of AmIS.

Previous studies have documented that a re-directed coastal current in front of Filchner-Ronne Ice Shelf (Hellmer et al., 2017; Naughten et al., 2021) and Ross Ice Shelf (Siahaan et al., 2022) drives a rapid warming of the ice shelf cavity in climate scenarios. The aim of this study is to determine whether the strength and direction of intrusive mCDW pathways in Prydz Bay remain the same or change in response to future climate change, and how these changes affect the melt experienced by AmIS.

This paper is structured as follows: Section 2 describes our regional model configuration, model forcing and experiments; Section 3.1 presents projections of AmIS basal melting and temperature on the continental shelf; Section 3.2 proposes a mechanism driving the increase in AmIS basal melting; and Section 4 presents conclusions and discusses the limits and implications of this work.

## 2 Models and Methods

### 2.1 Regional model configuration

The regional configuration used to study Amery Ice Shelf-Prydz Bay (hereafter AME025) is summarised below.

The ocean model used in this study is version 3.6 of the Nucleus for European Modelling of Ocean model (NEMO3.6, (Madec et al., 2017)). The configuration includes the physical ocean engine OPA (Madec et al., 2017), a sea-ice model LIM3 (Rousset et al., 2015) and an open ice-shelf cavity (Mathiot et al., 2017). The model domain extends $30°$ in longitude (60-90°E) and $10°$ in latitude (65-75°S) with nominal $0.25°$ horizontal resolution with grid spacing increasing from $\sim$7 km to $\sim$12 km with increasing distance from the South Pole. A 75-level $z^*$ vertical coordinate with a partial cell scheme is used in this work.

It is a non-linear free surface application allowing for variations of volume according to the vertical resolution, and can adjust at top and bottom cells to represent bathymetry more realistically. The thicknesses of grid cells range from 1 m at the surface to 200 m at about 6000 m. Following Global Ocean version 7 (GO7) provided by Storkey et al. (2018), and bathymetry is derived from ETOPO1 data set (Amante and Eakins, 2009) with GEBCO giving modifications in coastal regions (IOC et al., 2003). Bathymetry under the ice shelf is derived from IBSCO (Arndt et al., 2013) and the ice shelf draft is taken from BEDMAP2

(Fretwell et al., 2013).

The schemes and parameter values used in parameterisations are primarily based on a standalone global ocean configuration GO7 (Storkey et al., 2018), but the values of lateral diffusivity and viscosity have been changed according to the smallest grid spacing ($\sim$7 km) and time step (720s). Some physical schemes, such as the slip condition for the lateral boundary, from other regional modelling studies (Mathiot et al., 2017; Jourdain et al., 2017; Bull et al., 2021) are taken into account as well. A

55-term polynomial approximation of the Thermodynamic Equation of Seawater (TEOS-10, IOC et al. 2010) is used in our configuration (Roquet et al., 2015). A vector-form formulation of the momentum advection is applied. The vorticity term is computed using conserving potential entropy and horizontal kinetic energy (Arakawa and Lamb, 1981). Lateral diffusion of tracers is evaluated using Laplacian isoneutral mixing with a coefficient of 135 $m^2 \cdot s^{-1}$. Lateral diffusion of momentum uses bi-Laplacian geopotential viscosity with a coefficient of -1.08$\times10^{10}$ $m^4 \cdot s^{-1}$. The vertical eddy viscosity and diffusivity coef-

ficients are calculated from a Turbulent Kinetic Energy (TKE) scheme, with a background vertical eddy viscosity and vertical eddy diffusivity of $1.2\times10^{-4}$ $m^2 \cdot s^{-1}$ and of $2\times10^{-6}$ $m^2 \cdot s^{-1}$, respectively. The enhanced vertical diffusion parameterization is implemented for tracer convective processes with a coefficient of 10 $m^2 \cdot s^{-1}$. The non-linear bottom friction parameterization is chosen, with a non-dimensional bottom drag coefficient of $2.5\times10^{-3}$. The no-slip condition is implemented at the lateral momentum boundary.

Ice shelf thermodynamics are parameterised by the three-equation formulation with velocity dependent heat and salt exchange coefficients (Jenkins et al., 2010). The top boundary-layer thickness is set to 30 m below the ice shelf draft (or the first wet cell if the grid thickness is thicker than 30 m), and the top drag coefficient is set as $2.5\times10^{-3}$. The heat and salt exchange coefficients are $1.4\times10^{-2}$ and $4\times10^{-4}$, respectively. This implementation does not include external tidal forcing. The velocity of the tidal current at the top boundary is prescribed as 5 cm$\cdot$s$^{-1}$. A series of sensitivity experiments have been conducted to

determine the value of the prescribed tidal velocity (Figure S1).

## 2.2 UKESM1.0-LL forcing

We use climate projections made by the UK Earth System Model (UKESM1.0-LL) as part of the CMIP6 exercise (Sellar et al., 2020). UKESM1.0-LL is based on the HadGEM3-GC3.1 physical climate model, with additional atmospheric chemistry, and marine and terrestrial biogeochemistry components (Sellar et al., 2020). The ocean model for UKESM1.0-LL is NEMO3.6, and it is based on the 1° Global Ocean version 6 (GO6) configuration (Storkey et al., 2018), with 1° horizontal resolution and 75 vertical levels. The difference between GO6 and GO7 mentioned before is that GO7 is identical to a higher resolution GO6 0.25° configuration except there are open ice shelf cavities in GO7 (Storkey et al., 2018). The differences between the 1° and 0.25° GO6 configurations are the mixing and boundary conditions, which are adjusted according to time step and grid spacing (Storkey et al., 2018).

Model outputs from the first ensemble member (r1i1p1f2) of UKESM1.0-LL have been assessed by previous studies (Beadling et al., 2020; Heuzé, 2021; Purich and England, 2021; Roach et al., 2020; Bracegirdle et al., 2020; Meehl et al., 2020; Forster et al., 2020; Caillet et al., 2024). UKESM1.0-LL is well ranked in terms of various Southern Ocean and Antarctic sea properties (Caillet et al., 2024, Figure 1). To briefly summarise the evaluations of the modelled Southern Ocean in historical simulations, UKESM1.0-LL has relatively small biases in upper (0-100 m) ocean temperature (Beadling et al., 2020, Figure 6, Figure S3) and salinity (Beadling et al., 2020, Figure 6, Figure S4), and bottom temperature (Heuzé, 2021, Figure A3) and salinity (Heuzé, 2021, Figure A2) compared to other CMIP6 models. For the interior ocean, UKESM1.0-LL captures temperature and density structure across the continental shelf (Purich and England, 2021, Figure S2). It exhibits small cold (<-1°C) and fresh (<-0.15 psu) biases close to the shelf break along the 90°E longitudinal transect (Beadling et al., 2020, Figure S5) which is the easterly ocean boundary of the AME025 configuration. UKESM1.0-LL also exhibits a large internal climate variability of salinity averaged between 200 m and 700 m in front of AmIS, which is important for HSSW formation in Prydz Bay (Caillet et al., 2024, Figure 3c). In addition, UKESM1.0-LL shows accuracy in representing the position and strength of westerly jet over the Southern Ocean (Bracegirdle et al., 2020, Table 4). However, it is notable that UKESM1.0-LL has a higher climate sensitivity compared with other CMIP6 models (Forster et al., 2020, Figure 1; Meehl et al., 2020, Table 2), which might result in an unrealistically high surface warming (Forster et al., 2020). Additionally, UKESM1.0-LL overestimates the overall Antarctic sea ice loss in February (Beadling et al., 2020; Roach et al., 2020), which might cause a larger fresh bias at the ocean surface. However, it underestimates the February sea ice concentration in Prydz Bay (Roach et al., 2020, Figure S3), which may result in sea ice biases in our simulations. Moreover, UKESM1.0-LL has a relatively coarse 1° grid which cannot present some oceanographical features close to the Antarctic margin. Mathiot et al. (2011) suggest a minimum 0.5° nominal resolution to capture the Antarctic Slope Current and local gyres on the continental shelf.

## 2.3 Experimental design

Three experiments will be presented in this study: a historical simulation during 1976-2014 (Historical); a projection under SSP5-8.5 high emission scenario during 2015-2100; and a projection under SSP1-2.6 low emission scenario during 2015-2100.

Initial conditions consist of ocean temperature and salinity in January 1976 derived from GO7 (Storkey et al., 2018). The reason we chose GO7 rather than UKESM1.0-LL is that open ice shelf cavities are presented in G07 but not in UKESM1.0-LL. The oceanographic properties inside the AmIS cavity and in the open ocean are more physically consistent if the initial conditions are taken from one dataset.

Atmospheric and oceanic forcing of our simulations are provided by the r1i1p1f2 ensemble member of UKESM1.0-LL over the period of 1976-2100. The surface atmospheric forcing comprises daily variables, including near-surface air temperature at 2 m, specific humidity at 2 m, horizontal wind components at 10 m; surface downwelling longwave and shortwave radiation; and precipitation (rainfall and snowfall). It is applied through CORE bulk formulae (Large and Yeager, 2004). There is no freshwater or salt restoration in the regional domain. Note no external freshwater fluxes of surface runoff and iceberg calving are used in the AME025 configuration. These freshwater fluxes implicitly enter our domain through the lateral ocean boundary conditions, as they are in the UKESM1.0-LL outputs (Sellar et al., 2020). The lateral ocean boundary conditions consist of monthly ocean and sea ice variables, including sea surface temperature, sea surface height; ocean temperature, salinity, barotropic and baroclinic velocity; sea ice fraction, sea ice thickness and snow thickness.

The model outputs are saved as monthly mean. Therefore, the processes shorter than one month are not considered in this study.

The initialisation process is carried out by repeatedly simulating the first year (which is 1976) until the model drift for ice shelf melt rate, ocean temperature and salinity becomes small (Figure 2a, b). No smooth transition from December of a spin-up year to January of the next spin-up year is applied. There is physical discontinuity of the forcing between two spin-up years. Then a subsequent run is restarted from the last spin-up run with the continuous forcing from 1976 to 2100. The outputs of the subsequent run are used in the analysis.

The time series of area-averaged melt rates of AmIS in the spin-up run shows that it achieves equilibrium after about 10 years (Figure 2a). The melt rate dramatically declines from $\sim$16 m$\cdot$yr$^{-1}$ in the first spin-up year to $\sim$1 m$\cdot$yr$^{-1}$ in year 9 and becomes stable afterwards. The averaged temperature and salinity in the entire domain exhibit similar behaviours (Figure 2b).

Comparisons between the modelled melt rate in the Historical experiment (1976-2014) and the observational melt rate found in other studies (Wen et al., 2010; Yu et al., 2010; Depoorter et al., 2013; Rignot et al., 2013; Herraiz-Borreguero et al., 2016; Adusumilli et al., 2020; Rosevear et al., 2022a) are shown in Figure 2c. The time-mean melt rate during 1976-2014 in our study is 0.75 m$\cdot$yr$^{-1}$. This agrees with other estimates ranging from 0.5$\pm$0.12 to 1$\pm$0.4 (Figure 2c).

## 3   Results

### 3.1   The projected changes in the PB-AIS system by 2100

#### 3.1.1   The increased basal melting

Figure 3 shows the time evolution of the area-averaged AmIS basal melting from 1976 to 2100. The modelled melt rate is low and stable, followed by an abrupt increase in the 2060s under two scenarios (Figure 3a). Subsequently, the melt rate transitions

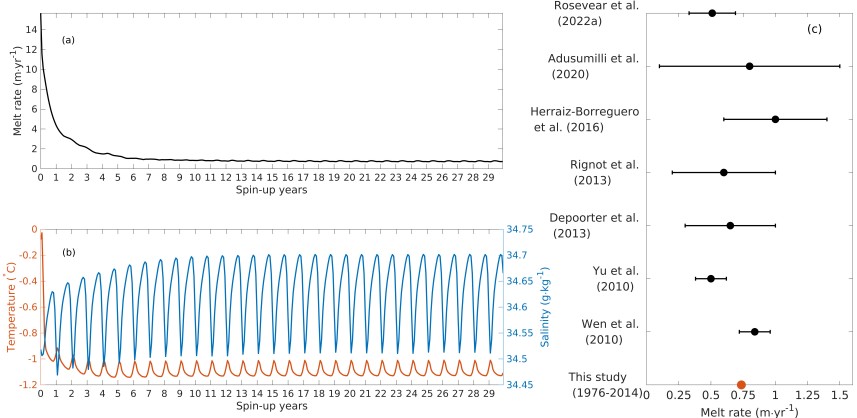

**Figure 2.** Time series of (a) area-averaged basal melting of AmIS ($\mathrm{m \cdot yr^{-1}}$), (b) temperature (°C, red line and y-axis on the left) and salinity ($\mathrm{g \cdot kg^{-1}}$, blue line and y-axis on the right) in 30-year spin-up run. (c) Comparisons of AmIS melt rate among different studies. The black dots represent the estimate of the AmIS melt rate and uncertainties from observational studies. The red dot shows the average of modelled AmIS melt rate from Historical simulation (1976-2014) in our study.

to a high melting state (Figure 3a). The modelled melt rate is $0.75 \pm 0.15 \ \mathrm{m \cdot yr^{-1}}$ (with 99.9% confidence intervals) over the period of 1976-2014. It is drastically increased to $13.14 \pm 2.36 \ \mathrm{m \cdot yr^{-1}}$ over the period of 2076-2100 under the SSP5-8.5 scenario or $8.10 \pm 1.53 \ \mathrm{m \cdot yr^{-1}}$ over the same period under the SSP1-2.6 scenario.

Figure 3b-d shows the spatial distributions of the AmIS basal melting during the historical, and before and after the increase. In general, after the increase in the 2060s the melting beneath AmIS strongly increases by multiple times and there is almost no

refreezing. The time-averaged melt rates over 1976-2014 illustrate that a high melting of $\sim 6 \ \mathrm{m \cdot yr^{-1}}$ occurs near the grounding line and most of the ice shelf experiences melting smaller than $2 \ \mathrm{m \cdot yr^{-1}}$ (Figure 3b). The freezing mainly occurs under the northwestern ice shelf with the highest value of $\sim 1.6 \ \mathrm{m \cdot yr^{-1}}$ near 69.5°E, -70.5°N (Figure 3b). Smaller freezing rates (<0.2 $\mathrm{m \cdot yr^{-1}}$) occur along the ice shelf edges (Figure 3b). This pattern fits the understanding of the buoyancy-driven meridional overturning circulation, in which warm and dense inflows downwell in the eastern ice shelf cavity while cold and fresh outflows

upwell in the western cavity. The time-averaged melt rates during 2075-2100 under SSP5-8.5 show that the melting occupies the entire ice shelf with the highest melt rates of $42 \ \mathrm{m \cdot yr^{-1}}$ near the grounding line (Figure 3c). Most of the areas experience melting between 15 and $35 \ \mathrm{m \cdot yr^{-1}}$ excluding the northeastern ice shelf which exhibits the relatively lower melting of $\sim 5$ $\mathrm{m \cdot yr^{-1}}$ (Figure 3c).

The time-averaged melt rates over 2015-2055 under SSP1-2.6 (Figure 3d) have similar behaviour to those under SSP5-8.5

(Figure 3c). The spatial distributions of melt rates over 2075-2100 for SSP1-2.6 are similar to those for SSP5-8.5, but with a smaller melt rate of $\sim 30 \ \mathrm{m \cdot yr^{-1}}$ near the grounding line and 10-25 $\mathrm{m \cdot yr^{-1}}$ over the central and the northwestern ice shelf (Figure 3d).

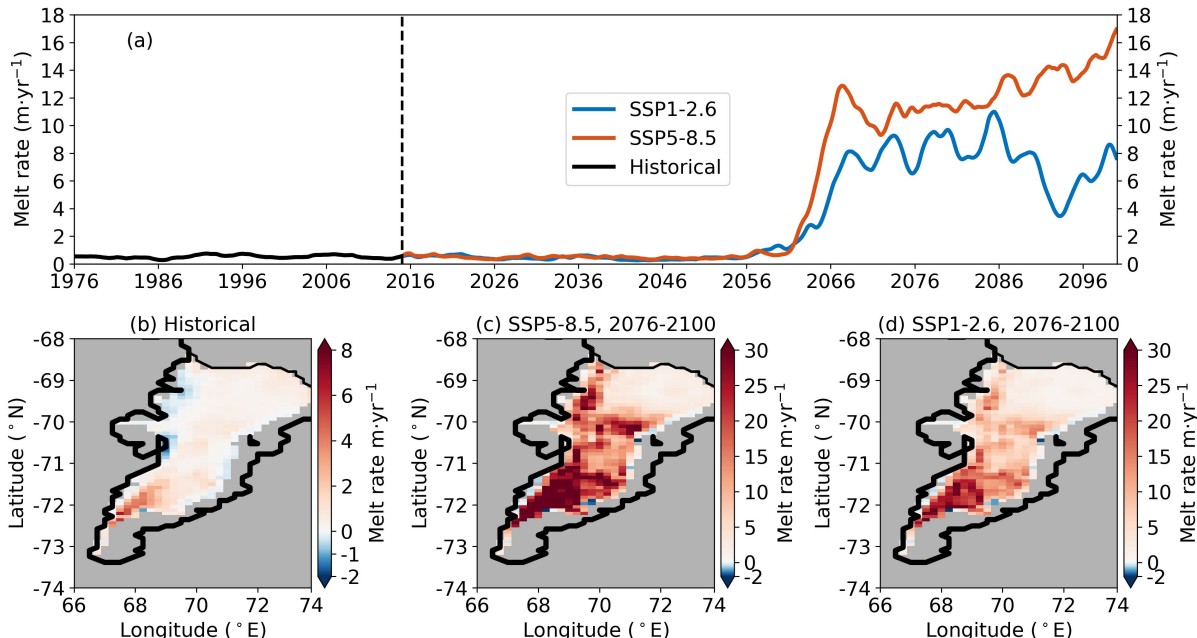

**Figure 3.** (a) Time series of the area-averaged melt rate $(m \cdot yr^{-1})$ from 1976 to 2100. A 12-month-running-average is applied. The dashed vertical line indicates the start of 2015. The time-averaged basal melting over the period of (b) 1976-2014 (Historical), (c) 2076-2100 under SSP5-8.5, (d) 2076-2100 under SSP1-2.6. The warm/cold colours represent melting/refreezing, respectively. Note the different colormap ranges. The colormap for (c) is saturated.

The melt rates in our simulations show higher sensitivity compared with other studies, for example, Naughten et al. (2018); Kusahara et al. (2023). This is inherited from the global UKESM1.0-LL model. The model forcing in Naughten et al. (2018) and Kusahara et al. (2023) is taken from the ACCESS-1.0 and MIROC-ESM, respectively. Meehl et al. (2020) and Forster et al. (2020) have quantified the climate sensitivity of these models and suggest UKESM1.0-LL has higher climate sensitivity than the other two models, which might result in a warmer ocean temperature (Sellar et al., 2020). This suggests that our regional model, forced by the outputs from the climate-sensitive UKESM1.0-LL, produces a stronger response to increasing emissions of greenhouse gases.

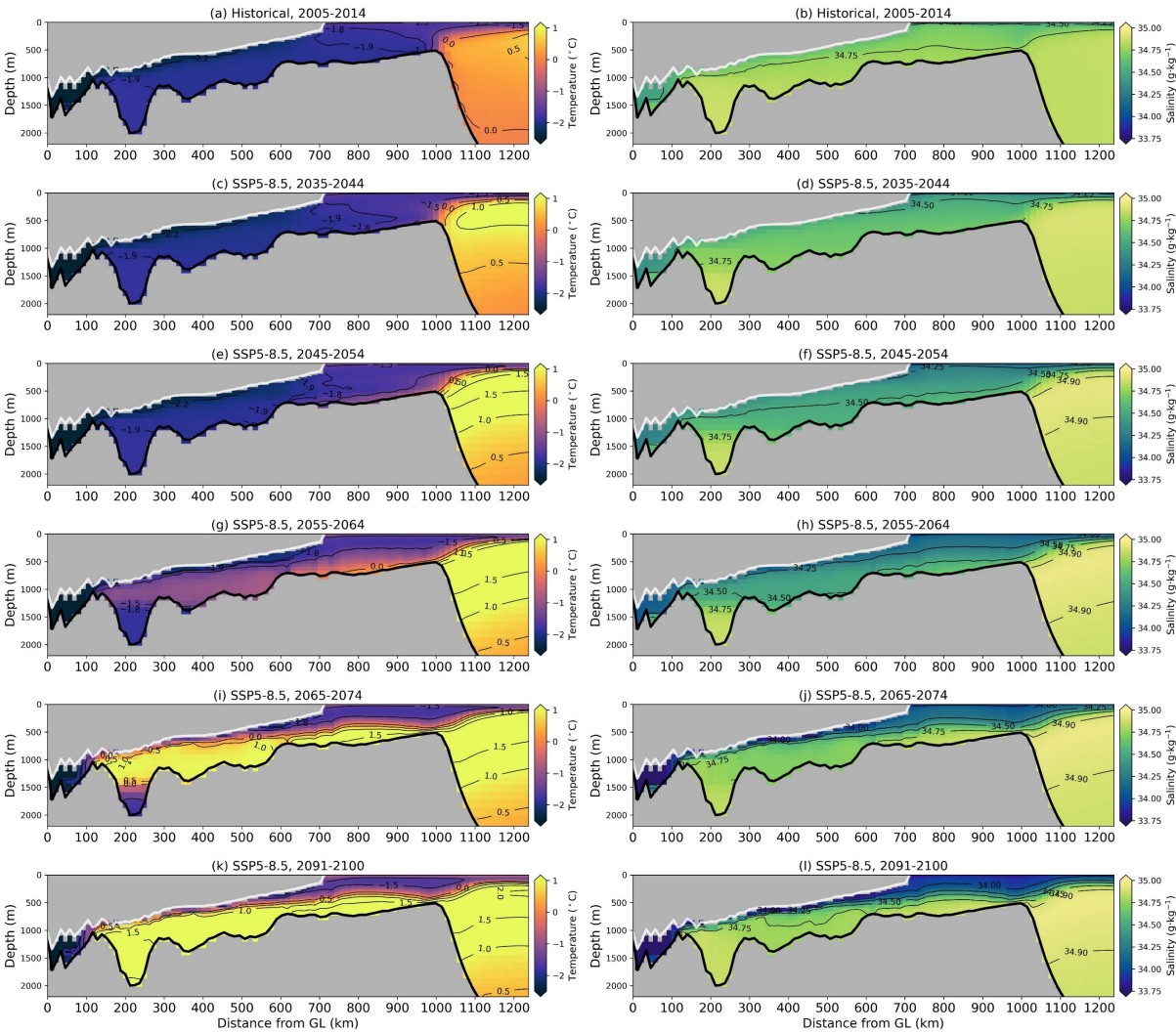

**Figure 4.** Time-mean of temperature, salinity at a transect extending from the grounding line (GL), Prydz Bay Gyre and the shelf break under SSP5-8.5 scenario. The coordinates show the distance from GL against the depth below sea level. The location of the transect is shown by the white line in Figure 1. The years of the time-mean are shown in the title of each panel.

### 3.1.2 Warming on the continental shelf

For the analyses below, the continental shelf and the shelf sea are defined by the areas between Antarctic Slope Current and the AmIS calving front/coastal line (Figure 1). The cavity is defined as the ocean beneath the AmIS draft.

There is a projected warming and freshening on the continental shelf and inside the AmIS cavity. Figure 4 shows temperature and salinity at a transect from the grounding line to the shelf break (the white line in Figure 1) under the SSP5-8.5 scenario. The warm mCDW off the slope is isolated at the shelf break during the years 2005-2014 (Figure 4a). The shelf sea is generally below -1.8°C at depths with a salinity of 34.5-34.75 g·kg$^{-1}$ between 2005 and 2014 (Figure 4a, b). The cavity temperature is below -1.9 °C and the salinity at the deeper cavity is higher than 34.75 g·kg$^{-1}$ (Figure 4a, b). This suggests that the cavity is filled by HSSW between 2005 and 2014. mCDW becomes warmer from the years 2035-2044 and penetrates across the shelf break, resulting in an increase in the shelf sea temperature to <-1.5 °C (Figure 4c). This is accompanied by a decrease in the shelf sea salinity to <34.5 g·kg$^{-1}$ (Figure 4d). The cavity temperature has little change with a slight freshening in the shallower depths (Figure 4c, d). From the years 2045-2054, mCDW on the seabed of the shelf sea spreads southward and intrudes into the cavity (Figure 4e), causing the cavity temperature increase to ~-1.5 °C during the years 2055-2064 (Figure 4g) and salinity decreases to 34.25-34.50 g·kg$^{-1}$ except for the deepest cavity (Figure 4h). This suggests that the dominant water mass in the cavity changes to mCDW between 2055 and 2064, consistent with the abrupt increase in melt rate in the 2060s. There is further warming in the entire area (Figure 4i, k). The cavity is flushed by even warmer mCDW of 0.5-1.5 °C during the years 2091-2100. The enhanced warming and melting of the ice shelf result in strong freshening in the upper cavity and the shelf sea, forming strong stratifications at depths of 300-500 m (Figure 4j, l).

The changes in water masses in the model domain under SSP5-8.5 are illustrated in Figure 5. The definition of water masses is given in Table 1. The water mass definitions are primarily based on Galton-Fenzi (2009) with modifications to adapt to our simulations. The plots show temperature against salinity on each grid cell inside the ice shelf cavity (the solid dots) and on the continental shelf (the open triangles). The black lines show the dilution relation when ice is melted by the warmest water mass in the cavity. During the years 2005-2014, there is a relatively colder mCDW on the continental shelf below 300m, which might induce the melting of the shallowest ice shelf (Figure 5a). HSSW occupies the cavity and controls the ice shelf melting at depths below about 800 m (Figure 5a). ISW clusters at the shallower depth of ~300 m inside the cavity (Figure 5a). ISW is also found between 800-2000 m following the Gade line with the source water of HSSW, suggesting this is the outflow of meltwater caused by HSSW (Figure 5a). The temperature of Antarctic Surface Water (AASW) is between -1 and -1.7 °C (Figure 5a). During the years 2035-2044 and 2045-2054, AASW becomes warmer and more stratified. The temperature of mCDW increases with the increase in the existence of mCDW in the cavity. HSSW still dominates the ice shelf melting (Figure 5b, c). However, the distributions of HSSW in the entire domain are less. HSSW becomes fresher and transforms to Low Salinity Shelf Water (Figure 5b, c). From the years 2055-2064, there is little HSSW found neither within the cavity nor on the continental shelf (Figure 5d-f). mCDW gradually occupies all depths below ~300 m inside and outside the cavity and controls the ice shelf melting (Figure 5d-f). Less ISW is formed in the shallower cavity, and it becomes fresher with potential density decreased from 27.6-27.8 kg·m$^{-3}$ during years 2005-2014 to < 27.2 kg·m$^{-3}$ from the years 2065-2074 onwards (Figure 5d-f). The meltwater on the Gade line, driven by the warm mCDW intruding the deepest cavity, becomes substantially warm (-2.5-1 °C) with depths from the years 2065-2074 (Figure 5d-f). In addition, the water mass in the shallower cavity (<500 m) is buoyant (26.6-27.8 kg·m$^{-3}$). This implies that the warmed and freshened outflow of meltwater increases the buoyancy inside

**Table 1.** Water mass definitions based on temperature (T, °C), salinity (S, g·kg$^{-1}$), potential density ($\rho$, kg·m$^{-3}$) and depth (m) characteristics. T$_f$ is the surface freezing point shown by the dashed line in Figure 5. AASW: Antarctic Surface Water; mCDW: modified Circumpolar Deep Water; HSSW: High Salinity Shelf Water; LSSW: Low Salinity Shelf Water; ISW: Ice Shelf Water.

| Water | T(°C) | S (g·kg$^{-1}$) | $\rho$ (kg·m$^{-3}$) | Depth (m) |
|-------|-------|------|------|------|
| AASW | T > T$_f$ | S < 34.0 | - | Depth < 200 |
| mCDW | T > -1.75 | - | $27.6 \leq \rho \geq 27.8$ | Depth $\geq$ 200 |
| HSSW | T$_f$ < T $\leq$ -1.75 | S $\geq$ 34.5 | - | - |
| LSSW | T$_f$ < T $\leq$ -1.75 | S < 34.5 | - | - |
| ISW | T < T$_f$ | - | - | - |

the cavity, enhancing the cavity circulations. AASW becomes warmer and its salinity and potential density spreads shrink from the years 2055-2064 onwards (Figure 5d-f), indicating that the stratification at the surface is enhancing.

The evolution of water masses suggests that the AmIS cavity starts to transition from a cold regime to a warm regime in the years 2055-2064 (Figure 5d) and eventually becomes a warm cavity from the years 2065-2074 (Figure 5e, f).

Time evolution of the depth-mean cavity temperature, salinity and the shelf sea temperature, salinity under SSP5-8.5 is shown in Figure 6. mCDW primarily occupies depths below 300 m in our simulations (Figure 5), so we choose a depth range of 300 m and below for the shelf sea properties to capture changes in mCDW. Figure 6a shows the timeseries of temperature. The shelf sea temperature begins to increase from about -1.5 °C in the 2030s to about 0.3 °C in the late 2060s, followed by a slower increase to above 0.5 °C in 2100 (Figure 6a). By contrast, the cavity temperature is stabilized at about -2 °C before the 2050s and then has an abrupt increase to about 0 °C in the late 2060s and a continuous increase to about 0.4 °C in 2100 (Figure 6a). The timeseries of the cavity temperature is consistent with that of the melt rate (Figure 3a). There is a roughly 20-year delay between the increase in the shelf sea temperature and the increase in the cavity temperature as well as the melt rate. The changes in salinity in the shelf sea salinity and the cavity are similar and more synchronised (Figure 6b). The two salinities fluctuate between 34.6 and 34.8 g·kg$^{-1}$ and start to decrease in the 2030s and then diverge in the 2040s when the cavity salinity has a larger decrease (Figure 6b). The difference is likely due to the reduction in sea ice production, which matters in dense water formation, while the increased mCDW is present on the continental shelf but has not yet reached the deeper cavity (Figure 5b, c). Reduced vertical convection resulting from a more stratified ocean may also cause the difference. Freshwater forcing from ice sheets is prescribed assuming the ice sheet mass is kept constant in UKESM1.0-LL (Sellar et al., 2020), which results in an additional snowfall on warmer ice sheets and increased freshwater released at the ocean surface. This bias is propagated through the lateral boundary to the regional domain. The shelf sea salinity and the cavity salinity have a slight bounce in the 2060s and are stable until 2100 (Figure 6b), suggesting a denser mCDW occupies the entire domain from the 2060s.

The delay of the cavity temperature increase and the larger freshening of the cavity salinity might suggest delayed processes connecting warming on the continental shelf and ice shelf melting such as the passage of warm water into the cavity.

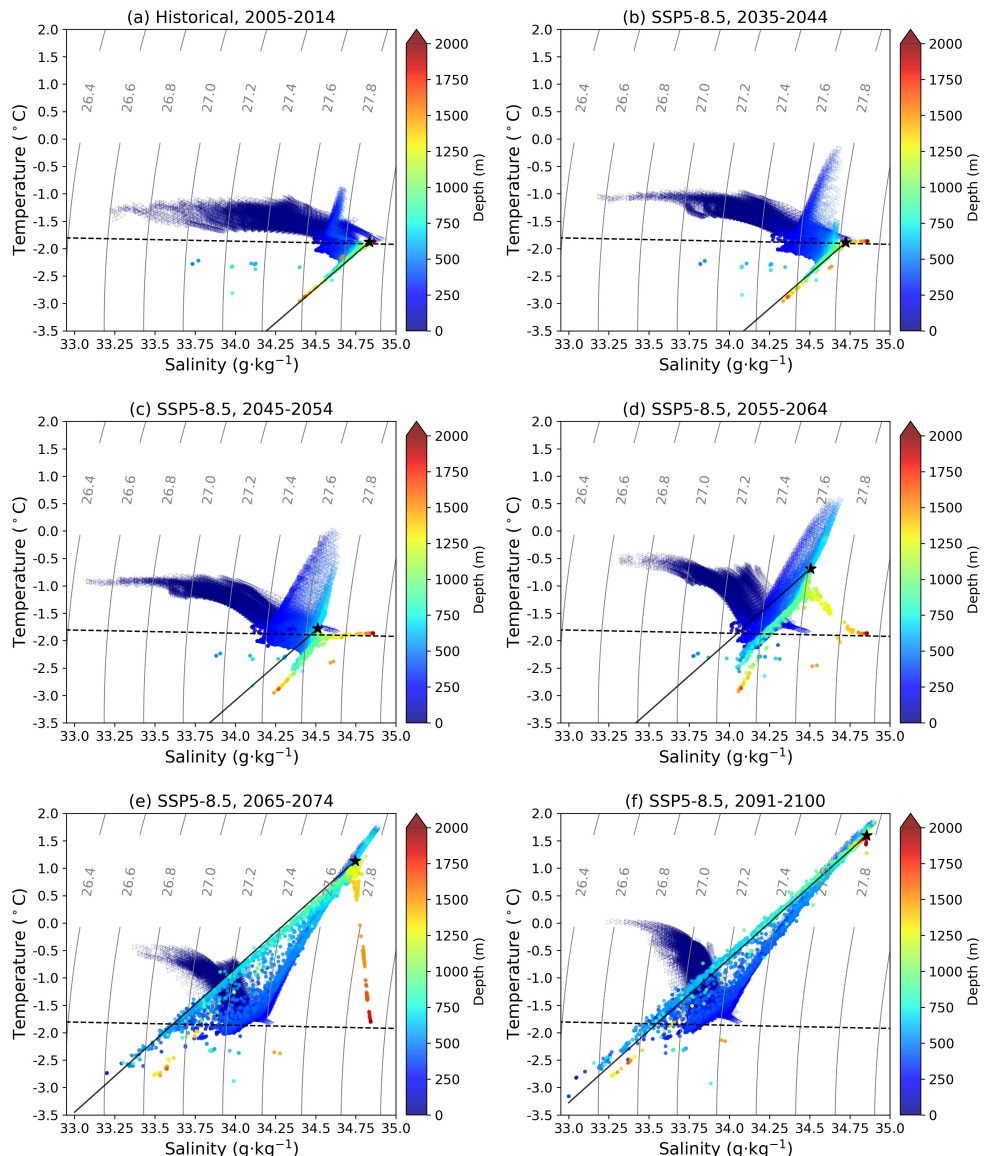

**Figure 5.** Time-mean of temperature against salinity diagram for each grid cell within the ice shelf cavity (the solid dots) and on the continental shelf (the open triangles) under the SSP5-8.5 scenario. The grey solid lines are potential density. The black dashed line shows the surface freezing point. The black solid line represents the Gade line, defined by end members of the warmest water mass in the cavity (the black pentacle). The colour schemes show the depth of each grid cell. The years of the time-mean are shown in the title of each panel.

The projected changes in temperature, salinity, water masses under SSP1-2.6 scenario have minor differences from those under SSP5-8.5 scenario (extra figures can be found in the supplementary document Section S2). The delay of the cavity temperature is found in the SSP1-2.6 simulations as well, with a smaller degree of warming and freshening.

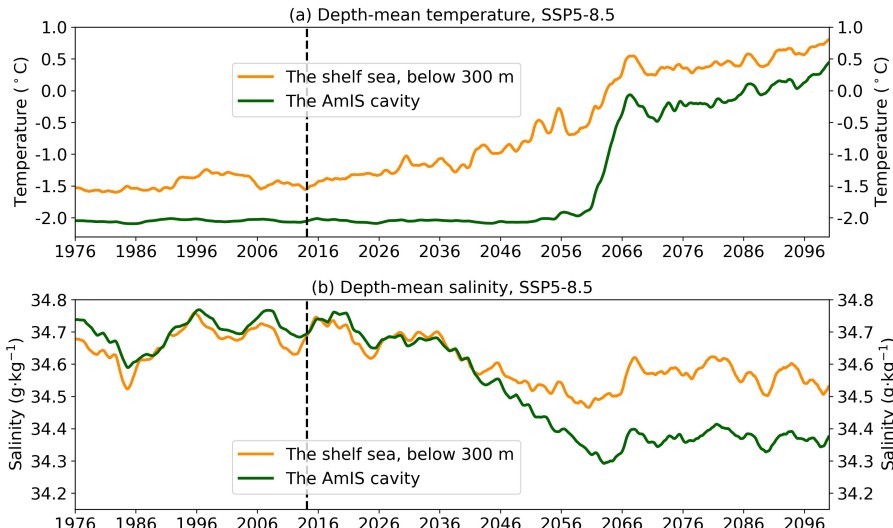

**Figure 6.** Timeseries of depth-mean (a) temperature and (b) salinity averaged within the ice shelf cavity (the green line) and on the continental shelf below 300 m (the orange line) under the SSP5-8.5 scenario. The vertical dashed line indicates the year of 2015. A 12 month running average is applied.

## 3.2 Mechanism causing the delayed increase in AmIS melting

### 3.2.1 Reversed current induced increase in heat into the cavity

Figure 7 shows the barotropic stream function (BSF) in the model domain under the SSP5-8.5 scenario. As the strength of BSF inside the cavity is much smaller than that outside the cavity, we present them with two different colour schemes. The red-blue colour schemes are for BSF on the open ocean and the orange-purple colour schemes are for BSF in the cavity. The positive/negative BSF represent anti-clockwise/clockwise circulation. The BSF of PBG is positive, suggesting PBG is anti-clockwise with a strength of ∼0.5 Sv and the PBG main flow (indicated by the solid purple line in Figure 7a) is off-shore during the years 2005-2014 (Figure 7a). The BSF near the AmIS calving front is positive with values of 0.1-0.2 Sv (Figure 7a), indicating one of the HSSW inflows is in the western ice shelf front during the years 2005-2024. The BSF in the deeper cavity is negative with values of -0.1 Sv during the years 2005-2024 (Figure 7a). This suggests that the cavity circulation is clockwise with an HSSW inflow in the eastern ice shelf front. During the years 2035-2044, 2045-2054, the positive and anti-clockwise PBG is gradually weakened, disappears and transitions to the negative BSF (Figure 7b, c). This is accompanied by a weakening of the cavity circulation (Figure 7b, c; Figure 8a), likely due to reduced formation of HSSW outside of the AmIS cavity (Figure 5b, c), which becomes less efficient at driving the cavity circulation. During the years 2055-2064, the negative BSF on the continental shelf is increased (Figure 7d) and the PBG strength is enhanced (Figure 8b), although the clockwise PBG has not yet been established (Figure 7d). The positive cavity BSF near the ice shelf front changes to negative, and the negative BSF in the deeper cavity is increased (Figure 7d) and the barotropic circulation in the cavity is strengthened during

the years 2055-2064 (Figure 8a). This suggests that the circulations driven by HSSW are very weak or non-existent, and the cavity circulation is controlled by mCDW. From the years 2065-2074, the clockwise PBG is well-established (Figure 7e, f) and the PBG strength is drastically increased (Figure 8b). The cavity circulation is greatly strengthened (Figure 7e, f; Figure 8a).

The evolution of the BSF distribution under SSP1-2.6 scenario behaves in a similar way to that under SSP5-8.5 scenario. However, there is a striking difference that the PBG and the cavity circulation strength decline in the 2090s in SSP1-2.6 while not in SSP5-8.5 (Figure 8). The decline is consistent with the decrease in the melt rate in the 2090s under SSP1-2.6 scenario. We will discuss the variations in Section 3.2.2.

To understand the changes in the relationship between ice shelf melting and ocean circulations inside and outside the cavity, we calculated the heat budget in the cavity as in Jourdain et al. (2017). Neglecting diffusion in ice and the interior ocean, the heat flux entering the ice cavity ($H_{in}$) is simplified into three components: the heat used to melt the ice shelf ($H_{lat}$), the heat that warms or cools the seawater within the cavity ($H_{HC\_VAR}$), and the remaining heat flux that exits the cavity ($H_{out}$):

$$H_{in} = H_{out} + H_{lat} + H_{HC\_VAR} \tag{1}$$

$$H_{in} = \rho_{ref} C_p \iint\limits_{r \in \text{front}, u > 0} u(r,z)(T(r,z) - T_f^0)\, dr\, dz \tag{2}$$

$$H_{out} = \rho_{ref} C_p \iint\limits_{r \in \text{front}, u < 0} u(r,z)(T(r,z) - T_f^0)\, dr\, dz \tag{3}$$

$$H_{lat} = L_f \iint\limits_{x,y \in \text{draft}} \rho_i m\, dx\, dy \tag{4}$$

$$H_{HC\_VAR} = \rho_{ref} C_p \iint\limits_{x,y,z \in \text{cavity}} \frac{dT(x,y,z)}{dt}\, dx\, dy\, dz \tag{5}$$

$\rho_{ref}$ is the reference seawater density of 1026 kg·m$^{-3}$, $C_p$ is the specific heat capacity of 3991.87 J·K$^{-1}$·kg$^{-1}$. $u(r,z)$ is the velocity perpendicular to the ice shelf front. The positive means velocity into the cavity. $T(r,z)$ is the temperature along the ice shelf front. $T_f^0$ is surface freezing point. $L_f$ equal to 334 J·kg$^{-1}$ is the latent heat of fusion of ice. $\rho_i m$ is the melt rate in unit of kg·m$^{-2}$s$^{-1}$. The latent heat of ice shelf ($H_{lat}$) is obtained from the model outputs. $dT(x,y,z)$ is temperature change on the same grid cell. $dt$ is one month. The variables used in the calculations are monthly mean model outputs.

Figure 9 shows time evolution of heat budget in the cavity under SSP5-8.5 and SSP1-2.6 scenarios. $H_{in}$ starts to increase in about 2040s and abruptly increases in the 2060s under two scenarios. This is coherent with the water mass changes in the cavity (Figure 5) and the development of the BSF of PBG (Figure 7). $H_{HC\_VAR}$ is at least one order magnitude smaller than $H_{lat}$ and $H_{out}$. The increase in $H_{lat}$ begins in the 2060s, and its behaviours are similar to the changes in the strength of the cavity circulation and PBG. This might suggest the increased ice shelf melting enhances the barotropic flow inside the cavity and in front of the ice shelf, as stronger pressure or density gradients are created (Jenkins, 2016; Jourdain et al., 2017). The melt rate-circulation strength coherence is similar to the melt-induced circulation found in the Jourdain et al. (2017) Amundsen Sea study. $H_{out}$ follows and is comparable to $H_{lat}$, suggesting that about 50% of heat is unused and leaves the cavity.

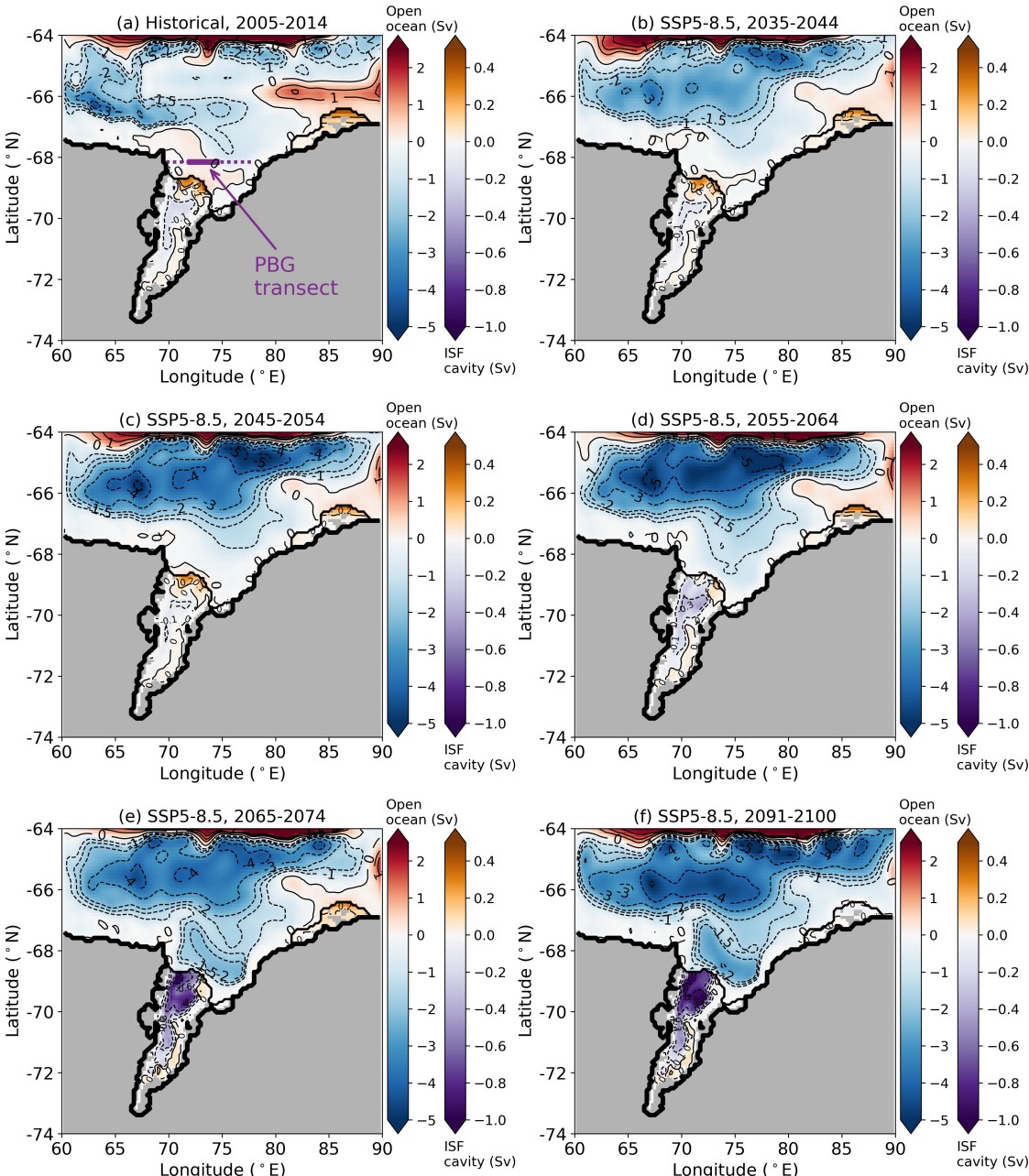

**Figure 7.** Time-mean of barotropic streamfunction (BSF, in units of Sv) under the SSP5-8.5 scenario. The red-blue colour schemes are for BSF on the open ocean. The values of BSF contours on the open ocean are -5, -4, -3, -2, -1.5, -1, 0, 1, 1.5, 2 Sv. The orange-purple colour schemes are for BSF in the ice shelf cavities. The values of BSF contours in the cavities -0.9, -0.6, -0.3, -0.1, 0, 0.1, 0.2 Sv. The positive/negative BSF represent anti-clockwise/clockwise circulation. The solid and dotted purple line shows the PBG transect and the extended transects to the coasts. The years of the time-mean are shown in the title of each panel.

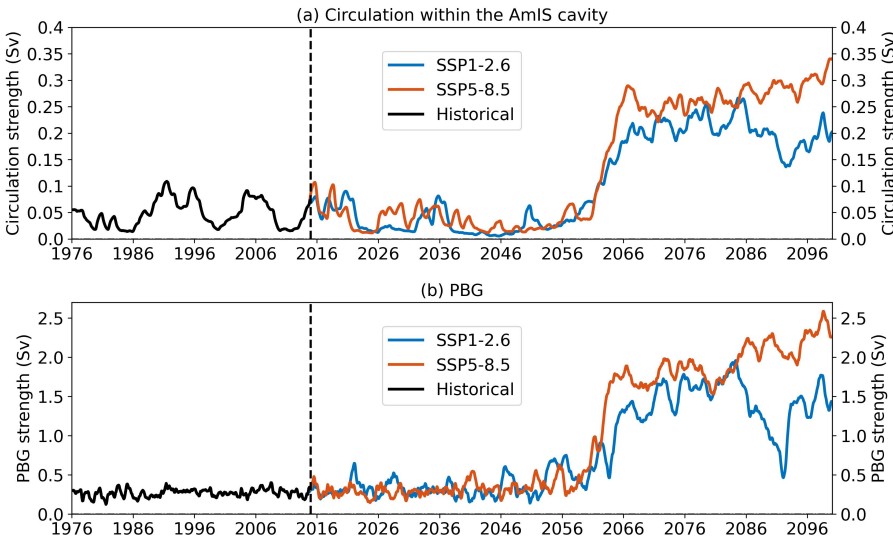

**Figure 8.** Timeseries of (a) the strength of the barotropic cavity circulation and (b) the strength of PBG. The strength of cavity circulation and PBG is defined as the absolute value of the area-mean barotropic stream function (BSF). The PBG area is defined using the closed BSF contour of -1.5 Sv in front of the ice shelf in Figure 7f. The vertical dashed line indicates the year of 2015. A 12 month running average is applied.

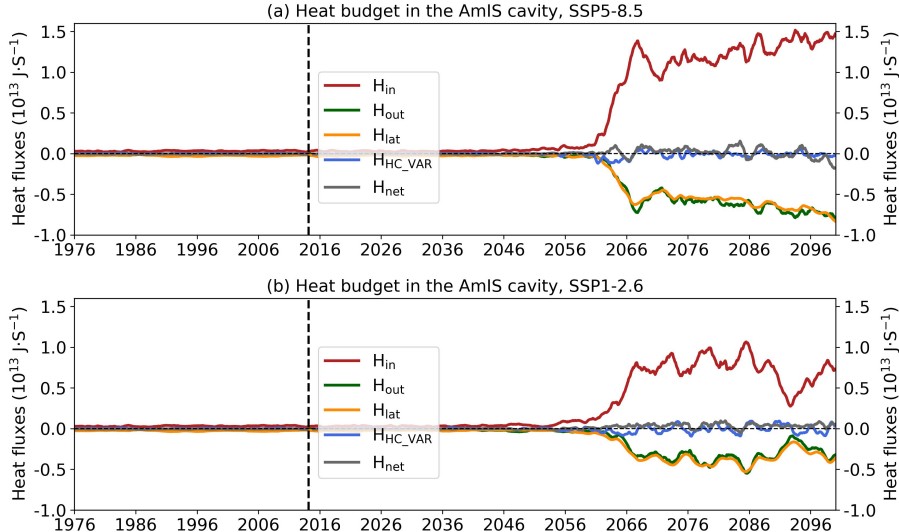

**Figure 9.** The heat budget of the ice shelf cavity for (a) the SSP5-8.5 scenario, (b) the SSP1-2.6 scenario. The heat of the ice shelf cavity is balanced by the heat flux into the cavity ($H_{in}$, the red line), the heat used to melt ice or latent heat of ice ($H_{lat}$, the orange line), the heat exit the cavity ($H_{out}$, the green line) and the heat used to warm/cool the seawater in the cavity ($H_{HC\_VAR}$, the navy line). The net heat flux ($H_{net}$) is equal to $H_{in}-(H_{lat} + H_{out} + H_{HC\_VAR})$. The vertical dashed line indicates the year of 2015. A 12 month running average is applied.

The changes in ocean circulations and heat fluxes in the cavity explain the question posed above. Why does the abrupt increase in the melt rate start ~20 years later than the increase in the shelf sea temperature? PBG is anticlockwise and weak before the 2050s, with off-shore transport at PBG transect (Figure 7a-c). Despite the warming and increased mCDW on the continental shelf (Figure 4a, c, e; Figure 5a-c), the processes of mCDW intruding the cavity are slow before the 2060s (Figure 8b; Figure 9). An effective pathway of mCDW into the cavity starts to establish in the years 2055-2064 (Figure 7d) and the clockwise PBG is well established and sustained from the years 2065-2074 (Figure 7e, f; Figure 8b), which transport massive heat into the cavity (Figure 9) and transforms the cavity to a warm regime (Figure 5e), resulting in the abrupt increase in melt rate after the 2060s (Figure 9).

### 3.2.2 Freshening-driven reversed current

The previous section demonstrated that the reversal of PBG main flow allows the increasing oceanic heat to penetrate the AmIS cavity, causing increased basal melting. We will now discuss what causes the PBG to reverse. The following analysis does not qualitatively vary between SSP5-8.5 and SSP1-2.6, so we only present the results from SSP5-8.5 here.

Here, we focus on a zonal section of PBG at the Amery Depression (the solid purple line in Figure 7a) and its extensions to the western and eastern bank (the dotted purple lines in Figure 7a). The area between the vertical dashed purple lines represents the PBG transect. Figure 10a shows that warming in the east of PBG begins in the 2040s and gradually spreads westward across PBG until the mid-2060s, when a sharp increase in temperature occurs. The temperature in the west of PBG is always cooler due to the injections of the cold outflow of meltwater (Figure 10a). The salinity exhibits notable changes (Figure 10b). There is a higher salinity and potential density in PBG and its western regions than in the east before the 2040s. Then the entire section has a decline in salinity until the mid-2060s, when the west of PBG experiences a sudden freshening, while PBG and its eastern section have an increase in salinity and potential density. This results in a reversed horizontal salinity and density gradient at the western boundary of PBG. Afterwards, the reversed gradient is sustained. As suggested in Jenkins (2016) and Jourdain et al. (2017), the changes in the barotropic coastal current induced by ice shelf melting can be a geostrophic adjustment due to the changes in pressure gradients. Here we present the sea surface height (SSH) gradient in Figure 10c. The SSH gradient mirrors salinity and density changes (Figure 10c). The positive/negative SSH gradient indicates that the SSH in the east is higher/lower than in the west. The SSH gradient is positive in PBG before the 2040s, suggesting there is a high SSH. The SSH gradient exhibits more fluctuations between the 2040s and 2060s, and a reversed SSH gradient in the west follows the density variations. This suggests that the outflow of AmIS meltwater may have a large impact on the local SSH gradients. The barotropic stream function (BSF) in the section tells a similar story that PBG is positive and anti-clockwise until completely reverses in the mid-2060s (Figure 10d).

Figure 11 shows timeseries of the main surface forcing under SSP5-8.5 scenario. Freshwater flux from ice shelf melting exhibits the same behaviours as the properties we analysed above (Figure 11a). The net sea ice production over the continental shelf shows a decrease from the 2040s (Figure 11b), which might be the source of the freshening of the PBG and the extended section in the 2040s (Figure 10b). The zonal and meridional surface stress at the ocean surface (either from wind or sea ice) does not behave in a similar way to the BSF of PBG (Figure 10c, d). The southward Ekman transport in the PBG section remains

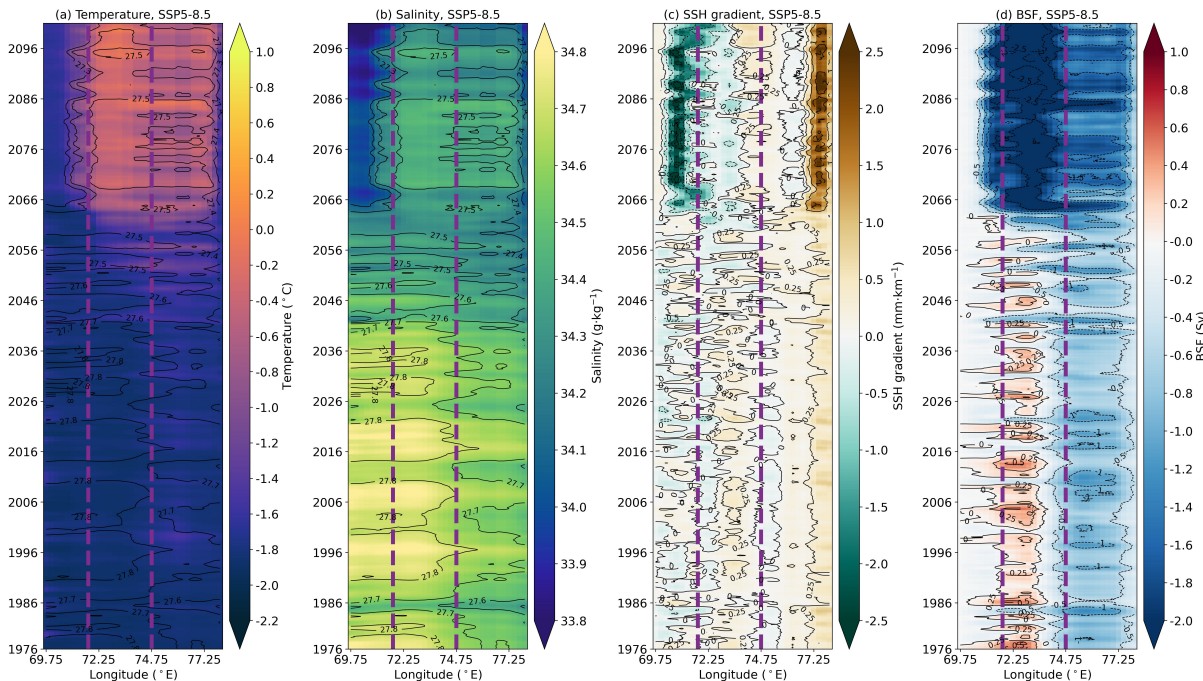

**Figure 10.** Hovmöller diagram of properties at the zonal transect shown in Figure 7a. (a) Depth-mean temperature below 300 m. (b) Depth-mean salinity below 300 m. The potential density below 300 m overlies temperature and salinity. The values of potential density contours are 27.3, 27.4, 27.5, 27.6, 27.7, 27.8 kg·m$^{-3}$. (c) Zonal sea surface height gradients (SSH gradient). A positive/negative SSH gradient indicates that the SSH in the east is higher/lower than in the west. The values of SSH gradient contours are -3, -2.5, -1.5, -0.5, 0, 0.25, 1, 2 mm·km$^{-1}$. (d) Barotropic streamfunction (BSF). The values of BSF contours are -2.5, -2, -1.5, -1, -0.5, 0, 0.25, 0.5, 1 Sv. A positive/negative BSF represents anti-clockwise/clockwise circulation. The transect in between the purple dashed lines is the PBG transect shown by the solid purple line in Figure 7a. A 12 month running average is applied.

at a magnitude of 0.2-0.3 Sv throughout the simulation. This is much smaller than the mean BSF of the reversed PBG which increases from 0.5 Sv in the 2060s to over 2 Sv thereafter (Figure 8). This suggests that surface stress is not a direct factor in the PBG reversal. The geostrophic component is probably more influential. However, Ekman pumping (calculated from the surface stresses at the ocean surface) in the PBG area, which strengthens the upwelling of mCDW onto the continental shelf (Greene et al., 2017), is enhanced between the 2040s and 2070s with large fluctuations afterwards (Figure 11e). The increased Ekman pumping coincides with the increased temperature in the PBG section (Figure 10a) and the reduction in sea ice production from the 2040s (Figure 11b). Figure 11 suggests that the enhanced Ekman pumping in the 2040s might be responsible for the freshening in the PBG section and the unstable PBG between the 2040s and 2060s, and the abruptly increased ice shelf melting further develops and sustains the reversed salinity/density gradient between PBG and its western regions. The Ekman pumping may also contribute to the changes in the local SSH gradients in the 2060s (Figure 10c).

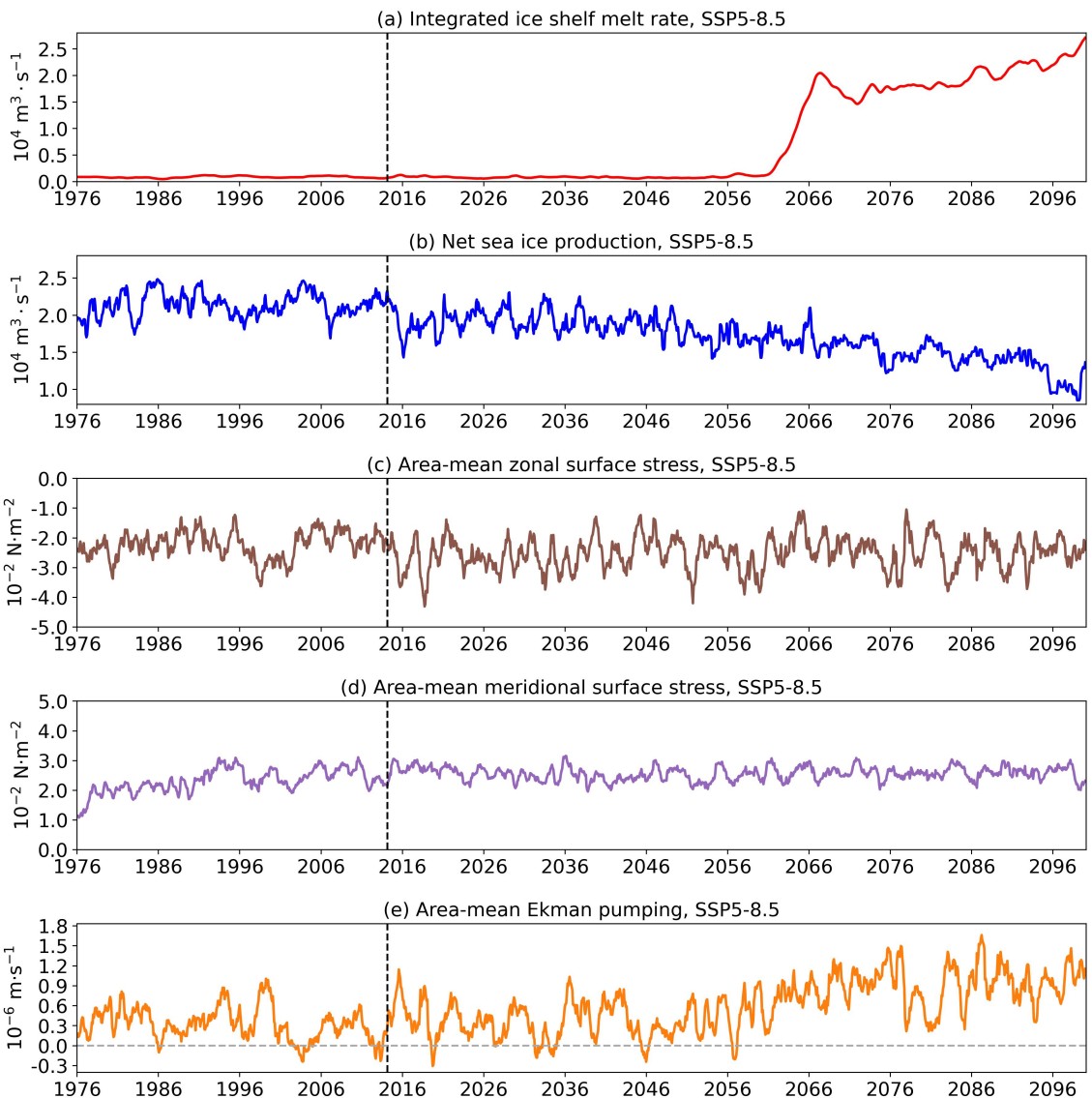

**Figure 11.** Timeseries of (a) integrated ice shelf freshwater fluxes ($10^4$ m$^3$·s$^{-1}$), (b) net sea ice production over the continental shelf ($10^4$ m$^3$·s$^{-1}$), (c) area-mean zonal surface stress within the PBG area ($10^{-2}$ N·m$^{-2}$), (d) area-mean meridional surface stress within the PBG area ($10^{-2}$ N·m$^{-2}$), (e) area-mean Ekman pumping within the PBG area ($10^{-6}$ m·s$^{-1}$) under the SSP5-8.5 scenario. The surface stress refers to the stress on the ocean surface (i.e. either from wind or sea ice). Ekman pumping is calculated based on the surface stresses. The PBG area is defined using the closed BSF contour of -1.5 Sv in front of the ice shelf in Figure 7f. The vertical dashed line indicates the year of 2015. A 12 month running average is applied.

Figure 10, 11 suggests that the reversal of PBG is controlled by the stronger freshening in the west of PBG due to the outflow of ice shelf meltwater, which caused a reversed density distribution and SSH gradient. PBG is reversed due to the adjustment of the new pressure gradient. Here we present a simple scale analysis of the meridional current. As the contribution from the surface stress $\tau_x$ is small (Figure 11c), the meridional current can be expressed as:

$$fv = \frac{1}{\rho_{ref}} \frac{dp}{dx}, \tag{6}$$

where $f$ = -1.4×10$^{-4}$ s$^{-1}$ is the Coriolis parameter, $v$ is time-averaged meridional velocity, $\rho_{ref}$ = 1026 kg·m$^{-3}$ is the reference seawater density, $dx$ the horizontal grid spacing. $dp$ is the horizontal pressure difference. Using the hydrostatic hypothesis,

$$dp = gHd\rho + g\rho_{ref}dSSH, \tag{7}$$

where $g$ = 9.81 m·s$^{-2}$ is gravity, $H \approx 600$ m is the depth to the seabed. $dSSH$ is the horizontal difference of sea surface height. Using a linear equation of state, and assuming that temperature difference do not produce a significant change in density (Figure 10a),

$$d\rho = bdS, \tag{8}$$

with $b$ = 0.78 kg·m$^{-3}$, $dS$ is the spatial salinity difference between PBG and its western regions. Therefore, the meridional velocity can be attributed to the effect of salinity gradient and sea surface height gradient:

$$v = \frac{g}{f} \left( \frac{Hb}{\rho_{ref}} \frac{dS}{dx} + \frac{dSSH}{dx} \right), \tag{9}$$

The estimates are shown in Figure A1. The term of salinity gradient ($\frac{Hb}{\rho_{ref}} \frac{dS}{dx}$) has a magnitude of 8×10$^{-6}$ (Figure A1a). The magnitude of the horizontal SSH gradient ($\frac{dSSH}{dx}$) is up to 4×10$^{-6}$ (Figure A1b), which is less than 50% of the salinity effect. The reconstructed meridional velocity based on Eq 9 agrees with the modelled velocity (Figure A1c). This analysis suggests that the effect of salinity gradient and SSH gradient compete (Figure A1a, b). However, the reversal of PBG is a consequence of the reversal of horizontal salinity differences between PBG and its western regions.

The reversal of PBG is also documented by Galton-Fenzi (2009). In his study, the clockwise PBG does not exist in winter because dense water formed in the Barrier polynya blocks the current from the Prydz Bay channel and prevents mCDW from accessing the ice shelf cavity (Galton-Fenzi, 2009). When the Barrier Polynya is active, the clockwise PBG is dumped (Galton-Fenzi, 2009). In our simulation, the seasonality of the PBG reversal is weaker than that in Galton-Fenzi (2009) (Figure 10d). This is due to overestimated sea ice concentration (SIC) in the Barrier polynya region (the white dashed box in Figure 12a) in our simulation (Figure 12), which is inherited from the overestimation of the summer SIC in the model forcing taken from the UKESM1.0-LL outputs (Roach et al., 2020). The modelled climatological summer SIC in the Barrier polynya between

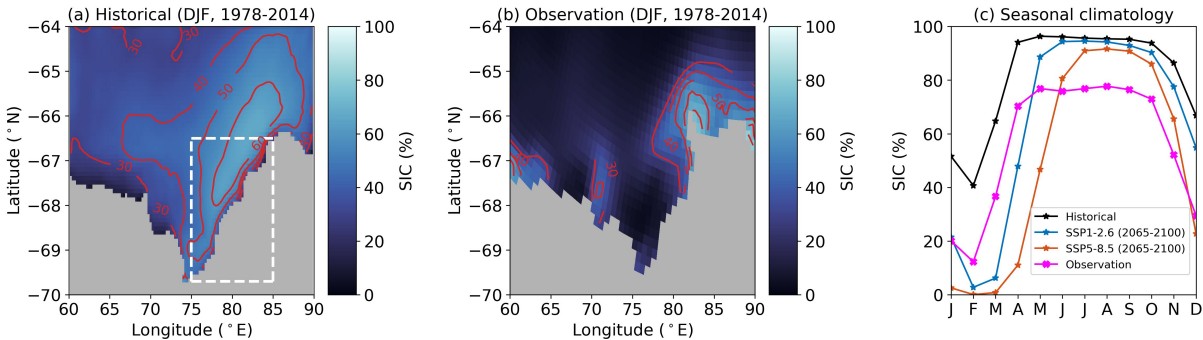

**Figure 12.** Time-mean summer (DJF) sea ice concentration (SIC) between 1978 and 2014 from (a) historical simulation in this study (Historical), (b) Observation. The observation dataset is Sea Ice Concentrations from Nimbus-7 SMMR and DMSP SSM/I-SSMIS Passive Microwave Data published by National Snow and Ice Data Center (NSIDC). The white dashed box in (a) shows the location of the Barrier Polynya. (c) Seasonal climatology of SIC within the Barrier Polynya. Seasonal climatology of two SSP scenarios is calculated between 2065 and 2100.

1978-2014 is 50-60% (Figure 12a), while the observed SIC in the Barrier polynya is up to 20% with 30-50% SIC in the upstream regions (Figure 12b). Our model failed to capture the spatial features. The seasonal climatology of SIC in the Barrier Polynya during the historical period shows that the modelled SIC in December, January and February is at least twice as high as the observation (Figure 12c). It also overestimates the SIC in other months. However, the SIC after December decreased

drastically in the two SSP scenarios, and the January, February and March SIC are below 20% (Figure 12c). This indicates that the overestimated SIC in the Barrier polynya weakens the seasonal reversal of PBG in our simulations. In addition, this also suggests that the overestimated sea ice mitigates the warm intrusions onto the shelf sea as the clockwise PBG is hard to establish well in summer. The reduction in sea ice production in the 2040s (Figure 11b) opens a wider gate to the formation of the clockwise PBG. Although sea ice in front of the ice shelf is not the direct trigger for the reversal of PBG in the 2060s,

it establishes the necessary preconditions for this event. This provides implications that a decrease in sea ice production in Prydz Bay could serve as a climate indicator of the increasing AmIS basal melt. Direct observation of ice shelf basal melting is challenging, but long-term sea ice records exist. A decreasing trend of sea ice may signal a forthcoming increase in basal melt rates.

The mechanism, in which the stronger freshening at the ice shelf front drives the reversal of PBG main flow, is valid under

385 the SSP1-2.6 scenario as well (Figure A2). The reduced sea ice in the Barrier polynya opens the channel and enables the establishment of the clockwise PBG. This mechanism is self-maintained: the warm water flushes the sub-ice shelf cavity, and the strengthened outflow of ice shelf meltwater reverses the salinity/density differences and stabilises the clockwise gyre, the southward main flow will carry more warm water to the cavity and sustain the feedback loop.

Here comes a remaining question: what causes the decline in melt rate (Figure 3a) and the associated weakening of PBG (Figure 8b) in the 2090s under SSP1-2.6 scenario. This is controlled by the variability of the shelf sea temperature in the upstream ocean boundary. Figure A3 shows timeseries of temperature and salinity on the continental shelf in the upstream boundary. There is a warming from 2015 to the 2040s-2050s, and temperature plateaus afterwards (Figure A3a). This trend has a good agreement with the shelf sea temperature changes (Figure 6a) and melt rate evolution (Figure 3a). This suggests that the local warming and freshening near the ice shelf front ultimately come from the upstream ocean boundary. In addition, there is a decrease in the shelf sea temperature in the upstream boundary from ~1.25 to 0.75°C in the 2090s under SSP1-2.6. This decline in temperature forcing results in the decrease in melt rate under SSP1-2.6 (Figure A3a) and a transient recovery of PBG (Figure A2d) in the 2090s. This implies that the gyre reversal is not irreversible. A decrease in temperature could stop this mechanism.

To validate the robustness of the freshening-driven mechanism, we conducted a second series of simulations with forcing taken from the r2 member of the UKESM1.0-LL ensemble. We chose the r2 member as its mid-depth temperature under the SSP1-2.6 scenario within the entire domain starts to increase $\sim 20$ years later than that in the r1 member (Figure S7c). The melt rate jump and PBG reversal under the SSP1-2.6 scenario also exhibit a delay compared to the r1 simulations (Figure S8-S10). A more detailed description of the r2 experiments can be found in Supplement Section S3. The differences between the r1 and r2 series of experiments highlight the importance of ocean boundary conditions, especially the variability of temperature forcing, to the ice shelf melt rate in our regional configuration.

## 4   Conclusions and discussions

This study investigates the future changes of AmIS-PB system under SSP5-8.5 and SSP1-2.6 scenarios. An abrupt increase in AmIS basal melting is projected to happen in the 2060s under both scenarios. The area averaged melt rate of AmIS is projected to increase from $0.75 \pm 0.15$ m·yr$^{-1}$ over the period of 1976-2014 to $8.10 \pm 1.53$ m·yr$^{-1}$ over the period of 2076-2100 under SSP1-2.6 and to $13.14 \pm 2.36$ m·yr$^{-1}$ over the period of 2076-2100 under SSP5-8.5. The time-mean melt rates during 2076-2100 (when it is in the high melting state) under both SSP1-2.6 and SSP5-8.5 display no refreezing beneath AmIS, and the melting at the grounding line exceeds 30 m·yr$^{-1}$. A drastic warming on the continental shelf and in the AmIS cavity causes the increased basal melting. However, the increase of temperature on the continental shelf is in the late 2030s, which happens ~20 years before the jump of AmIS basal melting in the 2060s. The delayed increase in AmIS melt rate is due to the reversal of PBG main flow.

PBG plays an important role in the changes in the basal melt of the AmIS. The clockwise PBG is not well established due to the sea ice in the active Barrier polynya, which obstructs its formation. The main flow is northward before ~the 2060s, preventing the increased mCDW from intruding into the AmIS cavity and delaying the increase in AmIS melting. However, the main flow of PBG reverses southward after the 2060s as the clockwise PBG is well established. The southward main flow imports substantial mCDW into the cavity, which leads to the increase of AmIS melting. The changes in PBG are due to: 1. The reduction in sea ice in the 2030s-2040s, which allows the establishment of the clockwise PBG. 2. The reversal of the horizontal

salinity (and then density) differences between the Amery Depression and its western regions. The redistribution of the salinity gradients is established by the strengthened outflow from the AmIS cavity after the 2060s.

A similar mechanism for the regime change is found in Filchner-Ronne Ice Shelf (Hellmer et al., 2012, 2017; Naughten et al., 2021; Siahaan et al., 2022) and Ross Ice Shelf (Siahaan et al., 2022). A redirection of coastal current is driven by a reversed density gradient across the Filchner Trough (Hellmer et al., 2012, 2017; Naughten et al., 2021; Siahaan et al., 2022)/ the little America Basin in the Ross Sea (Siahaan et al., 2022), which facilitates the penetration of mCDW into the ice shelf cavity and causes the regime change. The similar processes found in the sectors for the three large cold ice shelves emphasise the importance of buoyancy changes on the shelf sea in a warming future. This suggests the necessity of long-term records for the shelf sea salinity and sea ice in order to obtain an early warning of the regime change in the three large cold cavities.

Most cold ice shelf sectors have a structure with steep isopycnals at the continental shelf break (Thompson et al., 2018). The coastal geostrophic flow along the isopycnals is sensitive to the structure changes (Thompson et al., 2018). In this study, we only focus on the interactions between AmIS melting and local circulation. However, there remain many open questions for future studies. For example, how does the geostrophic flow respond to different components of freshwater fluxes from ice shelves, sea ice, icebergs, advections, precipitations, etc? What is the role ocean currents play in connecting the future changes in the freshwater components? What is the threshold of freshening on the continental shelf for the re-directed or reversed current? Does the threshold vary among different shelf sea sectors? A series of freshwater perturbation experiments across various climate scenarios would help address the above questions.

Quantifying the future stability of AmIS and its upstream ice sheets is beyond our research scoop, but this study can provide implications to some extent. Two previous ice sheet modelling studies (Pittard et al., 2017; Gong et al., 2014) conducted similar extreme experiments by applying enhanced basal melting of AmIS. Both studies suggested that only the collapse of almost the entire ice shelf by unrealistically high basal melting causes the grounding line to retreat beyond the topographic sill. Excluding the most extreme climate scenarios, AmIS attributes to sea level fall (Gong et al., 2014; Pittard et al., 2017). The stability of AmIS and the upstream ice sheets is primarily buttressed by the topographic sill tens of kilometres upstream of the grounding line (Gong et al., 2014; Pittard et al., 2017). In our SSP5-8.5 simulation, the basal melt rate is not as high as that applied in the extreme experiments in Pittard et al. (2017), with >50 m·yr$^{-1}$ and 100 m·yr$^{-1}$ for two extreme scenarios. It is reasonable to consider that AmIS will remain stable in the next century, however, the high basal melting puts the AmIS at risk of instability in the longer-term warming future. Coupling our standalone ocean configuration with an ice sheet model will be a future step to address the question of the AmIS stability.

We note the basal melting in this study might be overestimated. In the AME025 configuration, we use velocity-dependent "three-equation" parameterisation of ice-ocean thermodymanics (Jenkins et al., 2010). This parameterisation assumes that ice shelf melting is driven by the turbulent mixing due to the shear currents. However, the turbulent processes at the ice shelf-ocean interface can also be produced by the convection due to buoyancy forcing (Wells and Worster, 2008). For an ice shelf with a stable stratification at the ice shelf-ocean interface, such as AmIS (Rosevear et al., 2022b), the velocity-dependent "three-equation" parameterisation may overestimate basal melting of AmIS (Rosevear et al., 2022a). This results in a consequence that the projected melt rate is likely to be overestimated. In addition, UKESM1.0-LL has a higher climate sensitivity compared

with other CMIP6 models (Meehl et al., 2020; Forster et al., 2020) and previous generations of climate model (Sellar et al., 2020). This might produce an overestimated and more rapid warming in our regional projections compared with other studies (Naughten et al., 2018). Moreover, we do not include the frazil processes in this configuration. Galton-Fenzi et al. (2012)

suggests the inclusion of frazil in the AmIS simulation decreases the melt rates and increases the arrection rates. No frazil ice in our simulation may result in the overestimation. Another source of overestimation is the static ice shelf draft. Ice shelf melt rate is not only temperature dependent (Holland et al., 2008; Xu et al., 2013) but also basal slope dependent (Payne et al., 2007; Little et al., 2009; Magorrian and Wells, 2016). Steeper slopes might increase the heat entrained into the ice shelf and drive higher melting (Payne et al., 2007; Little et al., 2009; Magorrian and Wells, 2016). When AmIS is thinning, it will become

smoother and flatter, and the melt rate is expected to be to some extent decreased. Given the limited grounding line retreat of AmIS (Gong et al., 2014; Pittard et al., 2017), which feeds limited deep and steep grounded ice to the floating AmIS, therefore, the basal melting beneath the majority of AmIS north of the grounding line may be overestimated. Another limitation is that due to the relatively coarse grid spacing of ∼7-12 km for our model configuration relative to the estimated Rossby radius of 3 km over Prydz Bay (Liu et al., 2017), we cannot investigate the effect of mesoscale eddies on the AmIS basal melting. Liu

et al. (2017) suggested that 52% of the total onshore heat transport across a zonal transect (73-78°E, 67.5°S) in the Amery ice shelf front is induced by mesoscale eddies. Given the importance of mesoscale eddy on warm intrusion beneath Antarctic ice shelves (St-Laurent et al., 2013; Thompson et al., 2018; Stewart et al., 2018, 2019), it is worth employing a higher resolution model (∼1 km) to understand how mesoscale eddies impact future ice shelf melting.

The basal melt rate is projected to exceed 15 m·yr$^{-1}$ beneath the majority of AmIS after the abrupt increase under both

scenarios (Figure 3c, d). The thinning of ice shelf results in many changes, for instance, the geometry of the ice shelf cavity, the increased water column thickness under the ice shelf, etc. This is related to a scientific question: How does time-varying ice shelf draft modify ocean circulations in the cavity and ice shelf-ocean interactions in model simulations? Holland et al. (2023) suggests that a time-varying ice geometry of Thwaites Glacier leads to an increase in melting by more than 30% without any change in ocean forcing. However, we use a static ice shelf draft in the AME025 configuration, which limits the ability to

investigate such geometrical feedback. Future work would greatly benefit from the development of two-way coupled ocean-ice sheet models and more sophisticated Earth system models (Jordan et al., 2018; Smith et al., 2021; Siahaan et al., 2022; De Rydt and Naughten, 2023).

# Appendix A: Extra figures

## A1 The effect of salinity gradient and SSH gradient on PBG

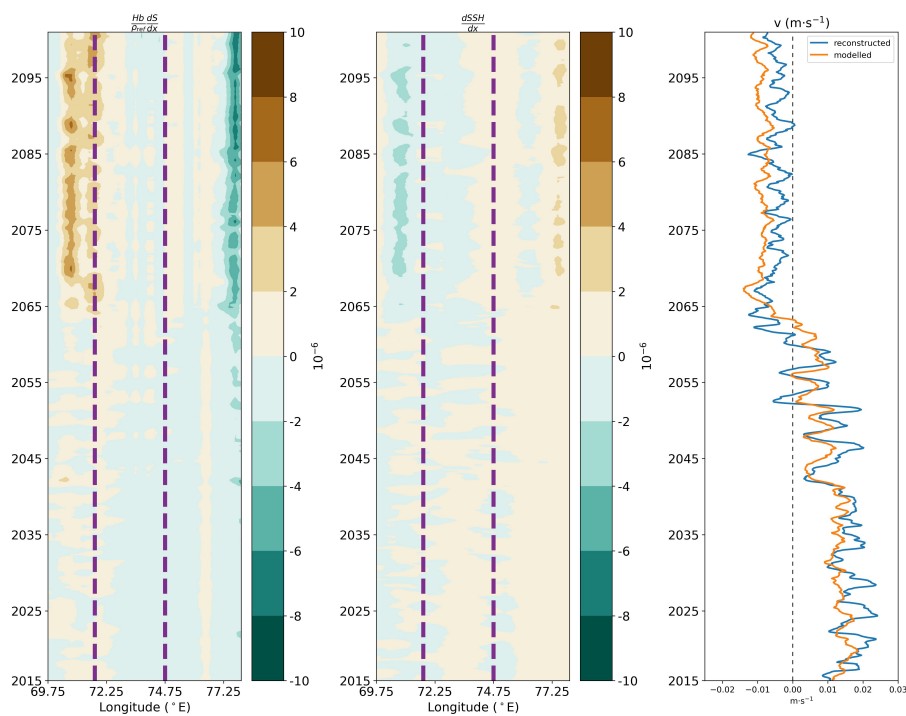

**Figure A1.** Hovmöller diagram of properties at the zonal transect shown in Figure 7a under the SSP-5.8 scenario for: (a) the salinity gradient term, (b) the sea surface height (SSH) gradient. A positive/negative salinity or SSH gradient indicates that the salinity or SSH in the east is higher/lower than in the west. (c) Timeseries of the averaged meridional velocity between the two dashed lines. The blue line shows the reconstructed velocity from Eq 9 and the orange line represents the modelled velocity. A 12 month running average is applied.

## A2 Extra figures for the SSP1-2.6 experiment

## A3 The forcing of temperature and salinity at the upstream ocean boundary

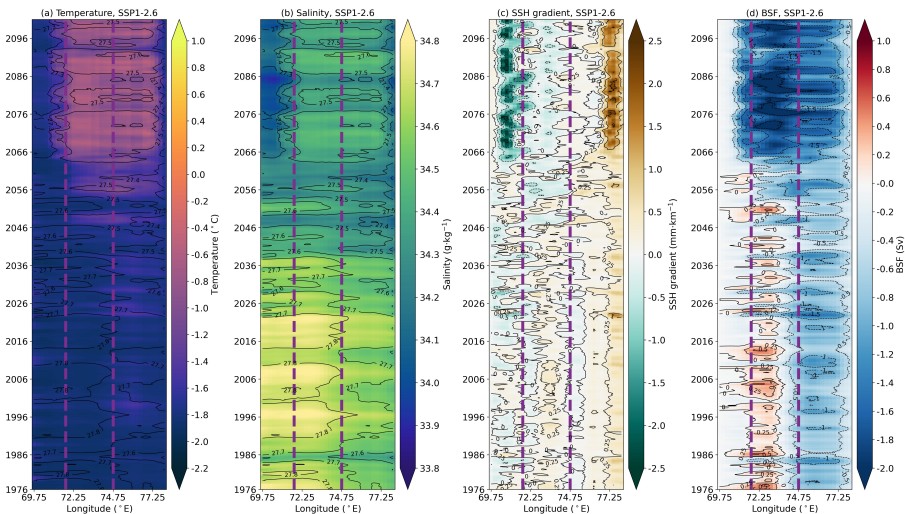

**Figure A2.** Hovmöller diagram of properties at the zonal transect shown in Figure 7a. (a) Depth-mean temperature below 300 m. (b) Depth-mean salinity below 300 m. The potential density below 300 m overlies temperature and salinity. The values of potential density contours are 27.3, 27.4, 27.5, 27.6, 27.7, 27.8 kg·m$^{-3}$. (c) Zonal sea surface height gradients (SSH gradient). A positive/negative SSH gradient indicates that the SSH in the east is higher/lower than in the west. The values of SSH gradient contours are -3, -2.5, -1.5, -0.5, 0, 0.25, 1, 2 mm·km$^{-1}$. (d) Barotropic streamfunction (BSF). The values of BSF contours are -2.5, -2, -1.5, -1, -0.5, 0, 0.25, 0.5, 1 Sv. A positive/negative BSF represents anti-clockwise/clockwise circulation. The transect in between the purple dashed lines is the PBG transect shown by the solid purple line in Figure 7a. A 12 month running average is applied.

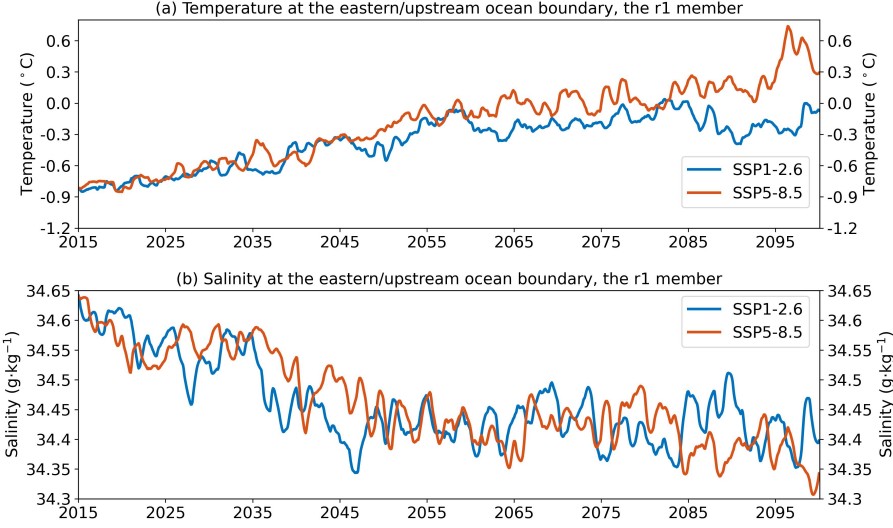

**Figure A3.** (a) Time series of the area-averaged temperature of ocean boundary forcing at the upstream boundary from 2015 to 2100. A 12-month-running-average is applied. (b) The same but for salinity ocean boundary forcing.

*Code and data availability.* The AME025 configuration can be obtained from https://zenodo.org/records/10797900. The UKESM forcing used in this study is free to download from the CMIP6 archive (Tang et al., 2019; Good et al., 2019a, b). The AME025 simulation outputs can be obtained from the corresponding author upon request.

*Author contributions.* JJ conducted the ocean simulations and prepared the manuscript. AP and CB helped to set up the regional model configuration and helped to interpret analysis. All authors commented on the manuscript.

*Competing interests.* The authors declare that they have no conflict of interest.

*Acknowledgements.* Jing Jin and Antony Payne were supported by Ocean Cryosphere Exchanges in ANtarctica: Impacts on Climate and the Earth system, OCEAN ICE, which is funded by the European Union, Horizon Europe Funding Programme for research and innovation under
495 grant agreement Nr. 101060452, 10.3030/101060452. OCEAN ICE contribution number 9. JingJin and Antony Payne were funded by UK Research and Innovation (UKRI) under the UK government's Horizon Europe funding Guarantee [grant number 10126975]. Christopher Bull was supported by the European Union's Horizon 2020 research and innovation programme under grant agreement no. 820575 (TiPACCs). The modelling work was carried out using the computational facilities of the Advanced Computing Research Centre, University of Bristol - http://www.bristol.ac.uk/acrc/.

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
