# Peer review of "Current reversal leads to regime change in Amery Ice Shelf cavity in the twenty-first century"

_EGUsphere, 2024_

## Author Comment (AC1)

**Responses to Reviewer 1**

We appreciate the time and effort Reviewer 1 has dedicated to providing a detailed review of this manuscript. Below we copy the referee comments in **bold black** and write our responses in **bold blue**. The changes to the manuscript are in *italic blue*.

**General comments**

**Overall, I find this to be a well written manuscript that looks at an interesting and topical case of whether, and how, the Amery ice shelf cavity could switch from a cold to a warm regime over the next 100 years. If so this would have implications for it's future stability and consequent contribution to sea level rise. I recommend publication with minor corrections provided my two main comments are addressed (one potentially not so minor).**

**1) My first comment is that the text in figures throughout the manuscript is far too small. Whether this is merely a formatting error in the cryosphere draft format or not I can not say, but it needs to be addressed before publication.**

**We thank the reviewer for the overall positive assessment and the suggestions for improvements. We have upgraded the figures. In order to address Reviewer 2's comments on Section 3, we have replaced the old figures in the preprint. We summarised the modifications here to help the reviewer follow the changes we have made. We also include our responses to Reviewer 2, and a new version of the manuscript and the manuscript with tracked changes so that the reviewer understands why we made these corrections.**

1 **We have added more discussion on the temperature changes and water mass transformations in Section 3.1.2. The additional analyses include:**

- **Replacing the depth-mean temperature (the old Figure 4) with temperature transect from the grounding line to the shelf break and beyond as suggested (Figure 4). The location of the transect is shown by the white line in Figure 1.**

- **Temperature against salinity diagrams for each ocean cell within and outside the ice shelf cavity (Figure 5) to illustrate the changes in water masses.**

- **Separated timeseries of depth-mean temperature/salinity within and outside the ice shelf cavity as suggested by Reviewer 1 (Figure 6).**

2 **We have additional analyses of the changes in ocean circulations within and outside the ice shelf cavity in Section 3.2.1. The additional analyses include:**

- Replacing the depth-averaged meridional velocity (the old Figure 5) with the barotropic stream function (Figure 7).

- Timeseries of the strength of circulations within and outside the ice shelf cavity (Figure 8).

3 We have replaced the analyses of PBG and PBECC heat transport (the old Figure 6) with the analyses of the heat budget in the ice shelf cavity (Figure 9).

4 We have additional discussion on the mechanism for the PBG reversal in Section 3.2.2. The modifications include:

- Replacing the analysis of the two boxes with a zonal transect of PBG (Figure 10).

- Analyses of the forcing of the gyre (Figure 11).

- Discussion on the impact of sea ice on the reversal of PBG (Figure 12).

**2) The second, and potentially larger comment, is the lack of discussion about how choices of ocean model parameters affect the conclusions. The runs as presented are just for two different climate scenarios, with no investigation into the effect of model parameters. Was sensitivity analysis carried out for viscosity and diffusivity, time step, grid resolution, (all properties known to have a large impact on modelled melt rates) etc? I do appreciate that ocean models are expensive to run and so all of this may not be feasible, but would like to see some discussion about what was done. I suspect that the qualitive conclusions will hold up, but would like to see some analysis backing this up.**

We appreciate the reviewer's understanding of the computational cost of running the ocean model. It takes two to three months to finish all simulations. This is impractical to test the sensitivity of the conclusions to model parameters, especially the model resolution. The sensitivity to the model set-up is very complex. This is another area of modelling study, which is out of our research scoop. However, if the reviewer is concerned about the robustness of the conclusions, we presented a second series of experiments which is forced by another UKESM1.0-LL ensemble (See supplement) to support the conclusions. In addition, we explained the reasons for choosing the parameters, for example, viscosity and diffusivity mentioned above, and other parameters of ice shelf thermodynamics which are more influential to the model results. The choices of parameters are questioned by Reviewer 2 as well.

We use a bilaplacian diffusion scheme in the AME025 configuration, the minimum stable requirement for the mixing coefficient ($A^h$) should meet

$$A^h < \frac{e^4}{64\Delta t}, \tag{1}$$

Where e is the smallest size of horizontal grids, $\Delta t$ is the timestep (Gurvan et al., 2017). In the AME025 configuration, the smallest grid size is 7 km and the timesteps are 720 s, which gives the mixing coefficient a limit of -5.02e+10 $m^4$/s. However, considering the very strong forcing of UKESM1.0 under SSP5-8.5, we would like to remove the potential risk of instability. The instability can be solved by reducing the timesteps or reducing the mixing coefficient. Reducing the

timesteps is computationally costly, so we decided to reduce the coefficient to -1.08e+10 m$^4$/s as in Bull et al. (2021). The values of viscosity and diffusivity have a minor impact on the modelled melt rate and ocean circulation. We tuned these parameters primarily for the model stabilisation.

A more important parameter is $U_t$, tidal velocity, in the ice shelf thermodynamics "three-equation" parameterisation (Holland and Jenkins, 1999). We use a velocity-dependent formula of the heat/salt exchange across the ice shelf-ocean interface (Jenkins et al., 2010):

$$\gamma_{T/S} = \sqrt{C_d}\Gamma_{T/S}U^*, \tag{2}$$

$$U^* = \sqrt{U_w^2 + U_t^2}, \tag{3}$$

where $C_d$ is the non-dimensional top drag coefficient of $2.5\times10^{-3}$, $\Gamma_{T/S}$ is non-dimensional heat and salt exchange coefficient of $1.4\times10^{-2}$ and $4\times10^{-4}$, respectively. $U_*$ is friction velocity which consist of $U_w$ averaged velocity in the top boundary layer (calculated by the model) and $U_t$ tidal velocity. $U_t$ is important for ice shelf melting as it affects the heat exchange between ocean and ice (Jourdain et al., 2019). It is common to employ external tidal forcing, for example Jourdain et al. (2017); Bull et al. (2021). However, it is impractical to obtain tidal forcing from UKESM1.0-LL, and we decided to prescribe tidal velocity as in Storkey et al. (2018) which provides the ocean core to UKESM1.0-LL.

The value of prescribed $U_t$ in our simulations was determined based on the results from a sensitivity experiment, in which we modelled the melt rate in 2006 by increasing the tidal velocity from 0 cm/s to 12 cm/s in increments of 2.5 cm/s. We have provided the experimental results in the supplementary document. The melt rate is increased with the increase in tidal velocity (Figure 1a). The modelled melt rate with tidal velocity of 2.5, 5 and 7.5 cm/s falls within the observational range (Figure 1b). To a different extent, the three runs overestimate the melt rate north of the grounding line and underestimate the melt rate at the grounding line. This is the limitation of our configuration. The relatively coarse resolution cannot represent the steep meridional distributions of basal melting. In addition, Rosevear et al. (2022) estimated the magnitude of typical tidal current under Amery Ice Shelf from four tidal constituents. The estimated magnitude is 9.8 cm/s, which yielded an annual-mean Amery melt rate of 1.1 m/yr in 2006. This is slightly higher than the observational estimate. Considering the tidal velocity of 2.5 cm/s is much smaller than the observed tidal velocity, and 7.5 cm/s might largely overestimate the future melt rate given a very high climate sensitivity of UKESM1.0, we decided to use 5 cm/s. We have included the results of the experiment tuning the tidal velocity in the supplementary document.

We did a second series of experiments with forcings taken from the r2 ensemble of UKESM1.0-LL to validate our conclusions. We chose the r2 ensemble because the ocean temperature under SSP1-2.6 for the r2 ensemble has a very different variability from that for the r1. The increase in ocean temperature in the Amery sector in the r2 ensemble happens 20 years later than the r1 ensemble, and it is followed by a larger decrease after the temperature peak. The r2 simulations show similar results of the transition from a cold to a warm cavity, the reversal of Prydz Bay Gyre, etc. The timing of the abrupt increase in the melt rate in the r2 experiment coincides with the temperature variability in the r2

[Figure]

**Figure 1.** (a) Melt rate against tidal velocity ($U_t$). (b) Comparison of the melt rate with different $U_t$ with the observational studies.

ensemble. We have supplied the results of the r2 simulations in the supplementary document.

**Line comments below.**

**L17 - (and throughout) Bathymetry as a positive number reads oddly to me. This may be a glaciology bias, but I think there would be less ambiguity to any glaciologists reading if in cases like this it is referred to as negative from sea level, or otherwise made explicitly clear what it is referring to.**

We thank the reviewer for providing a glaciological perspective. We have changed Bathymetry to the negative numbers in Figure 1. The vertical coordinates used in other figures are depth (pointing downwards) rather than height (pointing upwards). We keep the positive numbers and have claimed what the depth refers to in captions:

– **Figure 4**: *"... The coordinates show the distance from GL against the depth below sea level."*

– **Figure A1**: *"Hovemoller diagram (time against depth below sea level) of zonal-mean meridional velocity ..."*

**L75 - How does the ice shelf draft from BEDMAP2 compare to a more recent data set such as BEDMACHINE (Morlighem, 2022)?**

We compared the ice shelf draft from BEDMAP2 and BEDMACHINE (Figure 2). The ice shelf draft from BEDMAP2 and BEDMACHINE are not different on a large-scale (Figure 2a, b). There are some minor differences.

- Bedmap2 shows thicker ice near the ice shelf front (Figure 2c). Additionally, BEDMACHINE updated the ice shelf front. It outlines "Loose Tooth" while BEDMAP2 does not.

- Bedmap2 does not capture crevasses while BEDMACHINE does (Figure 2c).

- The ice at the grounding line is thinner in BEDMAP2 (Figure 2c). It is 300-1000 thinner than that in BEDMA-CHINE. BEDMAP2 also shows a less-defined channel near the southernmost grounding line.

BEDMACHINE has 500x500 m resolution, with two more additional sources to derive the ice thickness (Morlighem, 2022). By contrast, BEDMAP2 has a lower 1x1 km resolution. It captures less small geographical features, especially at the grounding line. However, the finest grid size in our model is about 7 km. It largely smooths these small-scale geographical features when interpolating the bathymetry/ice shelf draft data onto the ocean grids. Our future work is to update the bathymetry data to BEDMACHINE3 in a $1/12°$ configuration with a grid size of 1.5 km.

**L90 - To clarify, by "top boundary" I am assuming you are averaging ice-ocean boundary conditions for melting over 30m. How is this weighted if this occurs over more than one cell?**

Yes, it is the average over 30m below the ice shelf base or the first ocean cell below the ice shelf if the grid cell is thicker than 30m. It is weighted by grid cell thickness. For example, to obtain u velocity at the top boundary layer at a horizontal grid $u_{tbl}(i,j)$, let's assume three vertical levels are included in the top boundary layer, and the cell thickness for each grid on the U-point is $e3u(i,j,k)$, then the $u_{tbl}(i,j)$ is:

$$u_{tbl}(i,j) = \frac{\sum_{k=1}^{3} u(i,j,k) \cdot e3u(i,j,k)}{\sum_{k=1}^{3} e3u(i,j,k)} \tag{4}$$

**L136 - What time step is the model running at? Please include somewhere.**

Thanks for pointing out the missing information. The timestep is 720s. We have added to the manuscript.

**L202 - What is the exact value of Cp used?**

Cp is 3991.87 J kg/K. We have included this missing information.

**L253 - $\rho$ is used in equation 1 as a variable, not a constant. Suggest using $\rho_{ref}$ or similar here to refer to reference density or the like and avoid confusion.**

We thank the reviewer's suggestion. To address Review 2's question, we removed equation 1. However, we have changed $\rho_{ref}$ for the reference sea water density in other equations.

[Figure]

**Figure 2.** Thickness of Amery Ice Shelf from (a) Bedmap2, (b) BedMachineAntarctica-v3. (c) Comparison of ice shelf thickness between the two datasets. The BedMachineAntarctica-v3 data is interpolated onto the Bedmap2 grids. The positive/negative values represent that the ice shelf draft from Bedmap2 is thicker/thinner than that from BedMachineAntarctica-v3.

**L302 - To clarify, by net melt rate here are you referring to area averaged melt rate minus surface precipitation?**

We apologize for the ambiguous expression of "net melt rate". It is area averaged basal melt rate only. We will replace "net melt rate" with *"area averaged melt rate".*

**Fig 1- (and throughout) Is it possible to change the grey calving front lines to a more contrasting colour, as at present it blends into the contour lines?**

We thank the reviewer for the suggestions on improving the quality of figures. We have changed the calving front colour in Figure 1 to magenta. In Figure 3 and Figure 7, we increased the linewidth of the black coastal line to 4, and changed the calving front to a thinner black line (linewidth of 2).

**Fig 2- The text is too small to make out at the current figure size.**

We have increased the fontsize.

**Fig 2- The text is too small to make out at the current figure size.**

We have increased the fontsize.

**Fig 3 - The text, particularly in the legend, is too small to easily make out.**

We have enlarged the text.

**Fig 4 - Text is too small. Could average cavity temperatures be included to show the lag between shelf water temperatures and cavity temperatures?**

We removed this figure to address Reviewer 2's question. We have supplied figures with separated timeseries of the cavity temperature and the shelf sea temperature (Figure 6a).

**Fig 5 - Text by velocity arrow key too small. & Fig 8 - Legend text too small.**

We removed these figures to address Reviewer 2's question.

**Figure A4 - I find this a potentially misleading figure. I am assuming this is just the running total of increased melting compared to initial geometry? Without any feedback on the glacial system included in the model I am not sure how useful this is. In reality, a thinner ice shelf would flow faster, thus a 10m increase in melt rate would not necessarily lead to a 10m reduction in thickness per year at a given point.**

Yes, this figure is the running total of increased melting compared to initial geometry. We note this is an overestimate of ice shelf thickness reduction. The intention of the figure and the related context is to suggest the low potential of instability of Amery Ice Shelf and its upstream icesheets with our simulations. We have removed this figure and rephrased the context to avoid overinterpreting the results:

*"... In our SSP5-8.5 simulation, the basal melt rate is not as high as that applied in the extreme experiments in Pittard et al. (2017), with >50 m·yr$^{-1}$ and 100 m·yr$^{-1}$ for two extreme scenarios. It is reasonable to consider that AmIS will remain stable in the next century, however, the high basal melting puts the AmIS at risk of instability in the longer-term warming future. Coupling our standalone ocean configuration with an ice sheet model will be a future step to address the question of the AmIS stability."*

**References**

Bull, C. Y. S., Jenkins, A., Jourdain, N. C., Vaňková, I., Holland, P. R., Mathiot, P., Hausmann, U., and Sallée, J.-B.: Remote Control of Filchner-Ronne Ice Shelf Melt Rates by the Antarctic Slope Current, Journal of Geophysical Research: Oceans, 126, e2020JC016 550, https://doi.org/https://doi.org/10.1029/2020JC016550, e2020JC016550 2020JC016550, 2021.

Gurvan, M., Bourdallé-Badie, R., Bouttier, P.-A., Bricaud, C., Bruciaferri, D., Calvert, D., Chanut, J., Clementi, E., Coward, A., Delrosso, D., Ethé, C., Flavoni, S., Graham, T., Harle, J., Iovino, D., Lea, D., Lévy, C., Lovato, T., Martin, N., Masson, S., Mocavero, S., Paul, J., Rousset, C., Storkey, D., Storto, A., and Vancoppenolle, M.: NEMO ocean engine, https://doi.org/10.5281/zenodo.3248739, Fix broken cross-references, still revision 8625 from SVN repository., 2017.

Holland, D. M. and Jenkins, A.: Modeling Thermodynamic Ice–Ocean Interactions at the Base of an Ice Shelf, Journal of Physical Oceanography, 29, 1787 – 1800, https://doi.org/10.1175/1520-0485(1999)029<1787:MTIOIA>2.0.CO;2, 1999.

Jenkins, A., Nicholls, K. W., and Corr, H. F. J.: Observation and Parameterization of Ablation at the Base of Ronne Ice Shelf, Antarctica, Journal of Physical Oceanography, 40, 2298 – 2312, https://doi.org/10.1175/2010JPO4317.1, 2010.

Jourdain, N. C., Mathiot, P., Merino, N., Durand, G., Le Sommer, J., Spence, P., Dutrieux, P., and Madec, G.: Ocean circulation and sea-ice thinning induced by melting ice shelves in the Amundsen Sea, Journal of Geophysical Research: Oceans, 122, 2550–2573, https://doi.org/https://doi.org/10.1002/2016JC012509, 2017.

Jourdain, N. C., Molines, J.-M., Le Sommer, J., Mathiot, P., Chanut, J., de Lavergne, C., and Madec, G.: Simulating or prescribing the influence of tides on the Amundsen Sea ice shelves, 
[revised manuscript text omitted]

**Responses to Reviewer 2**

We thank Reviewer 2 for the constructive and thorough review of the manuscript. Below we copy the referee comments in **bold black** and write our responses in **bold blue**. The changes to the manuscript are in *italic blue*.

**General comments**

**This paper addresses the behavior of the Amery Ice Shelf under two climate change scenarios, SSP1-2.6 and SSP5-8.5. While the topic is indeed interesting, the paper falls short of meeting the minimum requirements for publication for several reasons, which I will detail below. The goal of this study is to estimate the future melt rate of the ice shelf by 2100 and to understand the mechanisms driving this melt. The authors utilize a regional simulation, AME025, forced with UKESM outputs. In my assessment, there are significant issues in every part of the paper, with the most critical problems found in section 3.2.**

**We are grateful to the reviewer for pointing out the problems of our study. We have made the corrections to address the major comments about Section 3. We summarised the modifications here, and the detailed changes can be found in the point-to-point responses.**

1. **We have added more discussion on the temperature changes and water mass transformations in Section 3.1.2. The additional analyses include:**

    - **Replacing the depth-mean temperature (the old Figure 4) with temperature transect from the grounding line to the shelf break and beyond as suggested (Figure 4). The location of the transect is shown by the white line in Figure 1.**

    - **Temperature against salinity diagrams for each ocean cell within and outside the ice shelf cavity (Figure 5) to illustrate the changes in water masses.**

    - **Separated timeseries of depth-mean temperature/salinity within and outside the ice shelf cavity as suggested by Reviewer 1 (Figure 6).**

2. **We have additional analyses of the changes in ocean circulations within and outside the ice shelf cavity in Section 3.2.1. The additional analyses include:**

    - **Replacing the depth-averaged meridional velocity (the old Figure 5) with the barotropic stream function (Figure 7).**

    - **Timeseries of the strength of circulations within and outside the ice shelf cavity (Figure 8).**

3. **We have replaced the analyses of PBG and PBECC heat transport (the old Figure 6) with the analyses of the heat budget in the ice shelf cavity (Figure 9).**

**4** We have additional discussion on the mechanism for the PBG reversal in Section 3.2.2. The modifications include:

- Replacing the analysis of the two boxes with a zonal transect of PBG (Figure 10).

- Analyses of the forcing of the gyre (Figure 11).

- Discussion on the impact of sea ice on the reversal of PBG (Figure 12).

We also included a new version of the manuscript with tracked changes at the end of this document.

**Specific comments**

**1) Section 2**

**There are several pieces of missing information:**

- **Which version of NEMO is used?**

  Thanks for pointing out the missing details. NEMO 3.6 is used in this study. We have added the version information in Section 2.1:

  *"The ocean model used in this study is version 3.6 of the Nucleus for European Modelling of Ocean model (NEMO3.6)."*

- **I understand that the configuration is an extraction of GO7 with some modifications to account for the specifics of a regional study. However, these modifications are not detailed in the text. For instance, at line 78, the authors mention that slip conditions are modified, but it is unclear how and why. What parameters are important here? Why not change the number of vertical levels as in Mathiot et al. (2023)?**

  Yes, the AME025 configuration is extracted from GO7. A free slip lateral momentum boundary condition for land and a partial slip condition for the Antarctic coast are used in GO7. However, GO7 is a global ocean configuration with open but non-interactive ice shelf cavities (Storkey et al., 2018). We changed the lateral momentum boundary condition to non-slip for land, coastal line and ice shelf interfaces in the light of other regional ice shelf studies such as Bull et al. (2021); Jourdain et al. (2017); Mathiot et al. (2017). In addition, the parameters of the ice shelf thermodynamics are considered primarily based on these ice shelf studies. It would be useful if the vertical levels could be increased to 121 levels. However, it is impractical in NEMO3.6. We have added the reasons why we changed the parameters in the manuscript:

  *"The schemes and parameter values used in parameterisations are primarily based on GO7 (Storkey et al., 2018), but the values of lateral diffusivity and viscosity have been changed according to the smallest grid spacing (~7 km) and time step (720s). Some physical schemes, such as the slip condition for the lateral boundary, from other regional modelling*

*studies (Mathiot et al., 2017; Jourdain et al., 2017; Bull et al., 2021) are taken into account as well."*

– **The viscosity and diffusion coefficients are set to -1.08e-10 m$^4$/s and 135 m$^2$/s, respectively. First, I suspect there is an error with the viscosity value, and it should be -1.08e+11; otherwise, it is extremely small. Second, these coefficients seem to be directly taken from the global configuration GO7 without adjustment, which is not appropriate. However, this might only marginally affect the simulations.**

We thank the reviewer for pointing out the typo. It is supposed to be -1.08e+10 m$^4$/s. We have corrected it. The viscosity coefficient in GO7 is -1.5e+11 m$^4$/s. We changed the coefficient based on the grid size and timestep of the AME025 configuration. For a bilaplacian diffusion scheme, the minimum stable requirement for the mixing coefficient (A$^h$) should meet

$$A^h < \frac{e^4}{64\Delta t}, \tag{1}$$

Where e is the smallest size of horizontal grids, $\Delta t$ is the timestep (Gurvan et al., 2017). In the AME025 configuration, the smallest grid size is about 7 km and the timesteps are 720 s, which gives the mixing coefficient a limit of -5.02e+10 m$^4$/s. However, considering the very strong forcing of UKESM1.0 under SSP5-8.5, we would like to remove the potential risk of instability. The instability can be solved by reducing the timesteps or reducing the mixing coefficient. Reducing the timesteps is computationally costly, so we decided to reduce the coefficient to -1.08e+10 m$^4$/s as in Bull et al. (2021).

– **The authors prescribe a constant tidal velocity of 5 cm/s. Where does this value come from? I suspect its impact on the melt is significant.**

Yes, the tidal velocity has a significant impact on the melt.

We did a sensitivity experiment to choose the value of 5 cm/s. We modelled the melt rate in 2006 with the tidal velocity increased from 0 cm/s to 12 cm/s in increments of 2.5 cm/s (Figure 1a). The melt rate is increased with the increase in tidal velocity. The modelled melt rate with tidal velocity of of 2.5, 5 and 7.5 cm/s falls within the observational range (Figure 1b). The three runs, to a different extent, overestimate the melt rate north of the grounding line and underestimate the melt rate at the grounding line. This is the limitation of our configuration. The relatively coarse resolution cannot represent the steep meridional distributions of basal melting.

In Rosevear et al. (2022), the estimated magnitude of typical tidal current under Amery Ice Shelf from four tidal constituents is 9.8 cm/s. The tidal velocity of 9.8 cm/s produced an annual-mean Amery melt rate of 1.1 m/yr in 2006, which is slightly higher than the observational studies. In addition, the melt rate with 9.8 cm/s tidal velocity failed to represent refreezing.

[Figure]

**Figure 1.** (a) Melt rate against tidal velocity ($U_t$). (b) Comparison of the melt rate with different $U_t$ with the observational studies.

Given the tidal velocity of 2.5 cm/s is much smaller than the observed tidal velocity, and 7.5 cm/s might largely overestimate the future melt rate given a very high climate sensitivity of UKESM1.0, we decided to use 5 cm/s.

We have mentioned the experiments in the manuscript and provided the results of the sensitivity experiment in a supplemental document:

*"A series of sensitivity experiments have been conducted to determine the value of the prescribed tidal velocity (Figure S1)."*

- **Is there any freshwater or salt restoring?**

  There is no freshwater or salt restoring in the regional domain. But the GO6 configuration, the ocean core of UKESM1.0, has restoring in the global domain. We have added this in the revised manuscript:

  *"There is no freshwater or salt restoring in the regional domain. However, the GO7 configuration—the ocean core of UKESM1—has restoring in the global domain (Storkey et al., 2018)."*

- **Lines 117 to 124: I do not understand the discussion about imposed melt in UKESM while AME025 has open interactive cavities.**

  Yes, the AME025 has open interactive cavities. However, we do not include freshwater fluxes of surface runoff and iceberg calving in our regional simulations. However, we do not include freshwater fluxes from surface runoff and iceberg calving in our regional simulations. While these are important components of surface freshwater fluxes, they may have a smaller impact on the Amery sector compared to ice shelf basal melting by the end of the 21st century (Siahaan et al., 2022; Coulon et al., 2024). We do not specifically employ these surface freshwater fluxes, but they implicitly enter our domain through the lateral boundaries, as they are prescribed in the global UKESM1.0 model.

  We have deleted the discussion about imposed melt in UKESM and rephrased the sentence:

  *"...No external freshwater fluxes of surface runoff and iceberg calving are used in the AME025 configuration. These freshwater fluxes implicitly enter our domain through the lateral ocean boundary conditions, as they are prescribed in the UKESM outputs (Sellar et al., 2020)."*

- **The spin-up phase involves repeatedly simulating the year 1976, but how is the transition between December and January handled? Is there not a discontinuity?**

  There is no transition handled. We note the limitation of this spin-up method. However, this method is efficient in reducing the model drift. We have added additional information about our spin-up method in Section 2.3:

  *"...No smooth transition from December of a spin-up year to January of the next spin-up year is applied. There is physical discontinuity of the forcing between two spin-up years. Then a subsequent run is restarted from the last spin-up run with the continuous forcing from 1976 to 2100. The outputs of the subsequent run are used in the analysis."*

**2) Section 3.1.1**

This section evaluates the melting rate by 2100. The authors focus on two periods: 2015-2055 and 2075-2100. However, it would be beneficial to explore the slow adjustment after the initial increase. For instance, there are fluctuations in the melt rate, especially in the 2090s for SSP1-2.6, that the authors did not explain. In my view, the most interesting aspect is the 10-year transition from a cold to a warm ice shelf around 2060-2070.

We briefly discussed the cause of the fluctuations in the melt rate in the 2090s under SSP1-2.6 scenario at Lines 294-297 in the preprint. This is due to a decrease in temperature on the continental shelf in the 2080s in the upstream/eastern ocean boundary conditions. All related processes in the 2080s-2090s under SSP1, such as a cooling on the continental shelf in the model domain, a recovery of PBG and a rebound in the melt rate, are ultimately controlled by the temperature changes in the upstream ocean boundary conditions.

We have done a double-check by running the projections with the r2 ensemble of UKESM1. We chose the r2 ensemble because the ocean temperature under SSP1-2.6 for the r2 ensemble has a very different variability from that for the r1. The increase in ocean temperature in the Amery sector in the r2 ensemble happens 20 years later than the r1 ensemble, and it is followed by a larger decrease after the temperature peak. The changes in the melt rate in the r2 experiment coincide with the temperature variability in the r2 ensemble. We have supplied this discussion and the results from the r2 experiment in a supplemental document.

Thanks for the suggestion on more investigations of the cold-to-warm transition. We have included more time-mean subpanels in the figures rather than two "before and after PBG reversal" subpanels in the revised manuscript. In addition, we have included more analyses on the time-evolution of sea ice, surface stress and circulations inside and outside the cavity to investigate the cold-to-warm transition.

Additionally, there is no discussion of estimates from other studies, which, I believe, found much weaker sensitivity. For instance, Kusahara et al. (2023) and Naughten et al. (2018) reported increases in melt by factors of 3 and 2, respectively, compared to a factor of 10 in this study.

We are not surprised by the much higher sensitivity in our projections, which is inherited from the global UKESM1.0-LL model. The forcing is MIROC-ESM in Kusahara et al. (2023) and ACCESS-1.0 and the multimodel-mean in Naughten et al. (2018). Meehl et al. (2020) have quantified the climate sensitivity of these models. For example, equilibrium climate sensitivity (ECS), which describes how much warming a model can produce with a doubling of $CO_2$ concentrations from its preindustrial level, is 3.8 K, 4.7 K and 5.3 K for MIROC-ESM, ACCESS-1.0 and UKESM1.0-LL, respectively. Another metric of model sensitivity – transient climate response (TCR) represents the response to increasing $CO_2$ – is 1.9 K, 2.2 K and 2.8 K for MIROC-ESM, ACCESS-1.0 and UKESM1.0-LL (Meehl et al., 2020). The higher climate sensitivity might result in a warmer ocean temperature (Sellar et al., 2020). This suggests our regional model, forced by the climate-sensitive UKESM1.0-LL, has a stronger response to climate warming.

**Siahaan et al. (2022) used a special version of UKESM1.0-LL which has open ice shelf cavities and is interactively coupled with an icesheet model (UKESM1.0-ice). In their projections, the ice shelf basal melt rate increases by a factor of 10. This is not entirely comparable with our regional projections, but it to some extent implies that the high sensitivity of the melt rate comes from the physical core of UKESM1.0-LL.**

**We have added the discussion about the higher sensitivity of melt rate in our simulations in Section 3.1.1:**

*"The melt rates in our simulations show higher sensitivity compared with other studies, for example, Naughten et al. (2018); Kusahara et al. (2023). This is inherited from the global UKESM1.0-LL model. The model forcing in Naughten et al. (2018) and Kusahara et al. (2023) is taken from the ACCESS-1.0 and MIROC-ESM, respectively. Meehl et al. (2020) and Forster et al. (2020) have quantified the climate sensitivity of these models and suggest UKESM1.0-LL has higher climate sensitivity than the other two models, which might result in a warmer ocean temperature (Sellar et al., 2020). This suggests our regional model, forced by the outputs from the climate-sensitive UKESM1.0-LL, has a stronger response to climate warming."*

**3) Section 3.1.2**

**This section evaluates the concomitant warming on the continental shelf. However, several issues and approximations need to be addressed.**

**The authors average temperatures in the 300-800 m depth range over the continental shelf without further explanation. It is necessary to specify the exact region considered and justify the choice of these depths.**

**We thank the reviewer for pointing out the missing information. We chose the 300-800m depth range as mCDW occupies the depth range on the continental shelf in our simulation (Figure 1-4). We aimed to show the mCDW changes on the continental shelf. We have added the reason why we chose the depth range in the revised manuscript:**

*"mCDW primarily occupies depths below 300 m in our simulations (Figure 5), so we choose a depth range of 300 m and below for the shelf sea properties to capture changes in mCDW."*

**Other modifications we have made is to replace "the depth range of 300-800m" with "below 300m". The deepest bathymetry on the continental shelf is about 800. The more accurate statement is "mCDW occupies from about 300 m to the seafloor" rather than "between 300 and 800 m". In addition, we have modified the definition of "the continental shelf". In the previous analyses, we used regions south of -67°N to represent the shelf sea. However, it is not accurate, and we have used bathymetry < 1000 m (The red arrow of the schematic ASC in Figure 1) to give a more precise geographical definition of the continental shelf in the revised manuscript. We have added a sentence to describe the**

**geographical definition:**

*"For the analyses below, the continental shelf and the shelf sea are defined by the areas between ASC and the AmIS calving front/coastal line (Figure 1). The cavity is defined as the ocean beneath the AmIS draft."*

**This averaging obscures the identification of the specific water masses that are warming. It would be more effective to examine sections extending from the grounding line to the shelf break and beyond. This approach would allow for a discussion on the warming or displacement of interfaces and the emergence of new water masses.**

We thank the reviewer for the suggestion on analysing water masses. We removed the figure of depth-averaged temperature (Figure 4 in preprint). Instead, we present a multi-panel figure to show different time-mean of temperature transect from the grounding line to the north of the shelf break under the SSP5-8.5 scenario as suggested (Figure 4 in the revised manuscript). In addition, we present a multi-panel T-S diagram to show the water mass changes on the continental shelf and within the ice shelf cavity (Figure 5 in the revised manuscript). In addition, we included the separated timeseries of temperature/salinity on the shelf sea and inside the cavity as suggested by Reviewer 1 (Figure 6 in the revised manuscript)). Section 3.1.2 has been rewritten as new analyses have been added. To save space we did not copy the new paragraphs here, but we attached the revised manuscript at the end. The results from the SSP1-2.6 simulation are also included in the supplementary document.

**The authors state that the temperature increase occurs several decades earlier than the increase in melt. However, based on Figure 4, the deviation from the steady state begins around 2040, approximately 15 years before the increase in melting, not decades. Additionally, the transition from the cold state to the warm state occurs clearly between 2060 and 2070 for both melting and warming, so the claim of a significant delay is questionable.**

Thanks for the reviewer to correct our language. We will clarify the specific description of the time scale and avoid using "decades".

The revised figures as suggested by the reviewers, which include the temperature transects from the grounding line to the shelf break and beyond (Figure 1, 2), water mass plots (Figure 3, 4) and the separated timeseries of temperature within and outside the cavity (Figure 5, 6), suggest the warming on the continental shelf starts in the 2030s even 20s, not 40s. The warming in the cavity begins in the 2050s, and the transition of the cavity starts in the 2060s. There may be some statistical argument for time differences between the warming outside of the cavity and the increase in melting, but we insist there is a delay.

**4) Section 3.2**

The goal of this section is to explain the changes in the melt rate. To do this, it is necessary to accurately depict the changes in water masses inside and ahead of the cavity, and their link to changes in ocean circulation. The analysis presented here is insufficient to address the question.

We are grateful for the reviewer's suggestions. We agree it is necessary to address the changes in water masses and their link to the changes in ocean circulation, and our analyses are insufficient. We have added additional analyses to address this question. We have made major corrections to Section 3.2 and rewritten this section. To save the reviewer's time, we summarised the changes in the analyses and the key finding here:

1. The changes in the water masses have been presented in Section 3.1.2. It shows that HSSW is decreased and the increased mCDW upwells on the continental shelf from the 2040s. The dominant water mass inside the cavity changes from HSSW to mCDW in the 2060s.

2. The spatial distributions of barotropic stream function inside and outside the cavity (Figure 7) and time evolution of the strength of the cavity circulation and PBG (Figure 8) have been presented in Section 3.2.1. This shows the reversal of PBG in the 2060s and the strength of PBG is increased afterwards. The strength of the cavity circulation is decreased in the 2040s as HSSW becomes inefficient to drive the cavity circulation, while the circulation strength is increased again in the 2060s as mCDW flushes the cavity due to the established clockwise PBG.

3. The heat budget of the cavity (Figure 9) has been added in Section 3.2.1. It shows the heat fluxes into the cavity start to increase in the 2040s. However, the massive heat, which changes the cavity regime, arrives in the 2060s as mCDW flushes the cavity.

The missing analyses are:

– **Temperature transects.**

We have provided an analysis of temperature transects (Figure 10a). Another modification is that the analysing region has been changed. The analysing region consisted of two boxes, one over PBG and the other over the western region of PBG. However, we showed the ocean circulation with barotropic stream function rather than meridional velocity as suggested by the reviewer. The two boxes will divide PBG, and this is not ideal to present an integral picture of the time-evolution of PBG. We changed the analysing region to a zonal transect of PBG with extensions to the western and eastern coastal lines (Figure 7a). As a consequence, we replaced the old Figure 7, which shows multi-panel plots of time-against-depth salinity in the two boxes, salinity differences between the two boxes and V velocity of PBG, with new multi-panel longitude-against-time plots (Figure 10). This new Figure 10 shows the temperature transects as requested by the reviewer, salinity, SSH gradients and barotropic stream function of PBG. The reflections of Figure 10 in the revised manuscript are not essentially different from the old

Figure 7 in the preprint. It also illustrates why and how PBG evolves. We have rewritten the analysis of the PBG transect from Lines 373-395 in the new version of the manuscript with tracked changes.

Another related modification is that we removed the old Figure 8 and the analysis of dynamic height anomaly from Lines 265 to 285 in the preprint. Dynamic height anomaly represents the pressure, which illustrates the reversal of PBG is due to new pressure gradients. However, this overlaps with the SSH gradients as shown in Figure 10c. Therefore, we removed the analysis of dynamic height anomaly to reduce the repetitive information. Additionally, the dynamic height anomalies are calculated offline, while the SSH variable is taken from model outputs, which is more convincing.

- **Analysis of mCDW transport inside the cavity with a proper closed budget.**
  We are unsure of the specific analysis of mCDW inside the cavity that the reviewer is expecting. For example, is it a budget as in Jourdain et al. (2017, eq. 6):

  $$\partial_t T = -u\partial_x T - v\partial_y T - w\partial_z T + D_l(T) + D_z(T) + l(T, z), \tag{2}$$

  which describes temperature changes due to advection, diffusion and surface fluxes. Or is it as in Jourdain et al. (2017, eq. D1), which describes heat flux into the cavity is used to melt ice, and warm/cool seawater in the cavity and the remaining heat flows out the cavity. Or is it an analysis of some processes else? We supplied an analysis as Jourdain et al. (2017, eq. D1) for two reasons. Firstly, the heat budget in the cavity is more illustrative of the regime change due to mCDW in the 2060s. We are aware this method cannot separate the heat transport from different water masses entering the cavity, which may not thoroughly address the reviewer's question. However, we have demonstrated the dominant water mass controlling the cavity changes from HSSW to mCDW in the new version of Section 3.1.2. It provides additional supporting information to the heat budget analysis. Second, we did not save model outputs shorter than monthly, and we did not output the diffusion variables. The offline calculations of Jourdain et al. (2017, eq. 6), especially the diffusion terms, are not ideal. We have to re-run all experiments to obtain those data, which needs an additional two or three months. To shorten the reviewing process, we present the analysis of the heat budget as Jourdain et al. (2017, eq. D1), and we appreciate the understanding from the reviewer.

  Following Jourdain et al. (2017, eq. D1), neglecting diffusion in ice and the interior ocean, the heat flux entering the ice cavity ($H_{in}$) is simplified into three components: the heat used to melt the ice shelf ($H_{lat}$), the heat that

warms or cools the seawater within the cavity ($H_{HC\_VAR}$), and the remaining heat flux that exits the cavity ($H_{out}$):

$$H_{in} = H_{out} + H_{lat} + H_{HC\_VAR} \tag{3}$$

$$H_{in} = \rho_{ref} C_p \iint\limits_{r \in \text{front}, u>0} u(r,z)(T(r,z) - T_f^0)\, dr\, dz \tag{4}$$

$$H_{out} = \rho_{ref} C_p \iint\limits_{r \in \text{front}, u<0} u(r,z)(T(r,z) - T_f^0)\, dr\, dz \tag{5}$$

$$H_{lat} = L_f \iint\limits_{x,y \in \text{draft}} \rho_i m\, dx\, dy \tag{6}$$

$$H_{HC\_VAR} = \rho_{ref} C_p \iint\limits_{x,y,z \in \text{cavity}} \frac{dT(x,y,z)}{dt}\, dx\, dy\, dz \tag{7}$$

$\rho_{ref}$ is the reference seawater density of 1026 kg·m$^{-3}$, $C_p$ is the specific heat capacity of 3991.87 J·K$^{-1}$. $u(r,z)$ is the velocity perpendicular to the ice shelf front. The positive means velocity into the cavity. $T(r,z)$ is the temperature along the ice shelf front. $T_f^0$ is surface freezing point. $L_f$ equal to 334 is the latent heat of fusion of ice. $\rho_i m$ is the melt rate in unit of kg·m$^{-2}$s$^{-1}$. The latent heat of ice shelf ($H_{lat}$) is obtained from the model outputs. $dT(x,y,z)$ is temperature change on the same grid cell. $dt$ is one month. The variables used in the calculations are monthly mean model outputs. The timeseries of the heat budget components show a sudden increase in $H_{in}$ and $H_{lat}$ in the 2060s, suggesting the sudden flush of mCDW causes the increased melting as expected (Figure 9).

– **Use of daily outputs instead of monthly, since $(v \cdot T)^{1mo} \neq v^{1mo} \cdot T^{1mo}$ especially given the strong variability in this area.**

Yes, we agree the use of daily data is more helpful in investigating the variability of PBG. However, we saved monthly variables only to save data storage space. We will be short of space if we save daily outputs for the runs over 200 years in total. We have to re-run simulations to obtain the daily outputs. Additionally, the university's HPC was upgraded this summer, and the modules, particularly the compiler used in this study, are no longer available. Guarino et al. (2020) suggest the impact of HPC machines on the reproducibility of the climate simulations. Although this has a low likelihood of changing the main conclusions, we have to re-run monthly outputs as well to ensure that the new simulations do not have significant differences from the ones we used in the preprint. However, this will take two to three months to finish all simulations, and it slows down the reviewing processes.

An alternative solution is to remove the analysis of meridional heat transport for the PBG and PBECC. Instead, we present the analysis of the heat budget in the cavity as we replied previously. The analysis of meridional heat transport of PBG and PBECC cannot quantify how much heat outside the cavity is transported into the cavity. While it shows increased onshore heat during the 2060s due to the reversal of the PBG, this is somewhat redundant as we have already demonstrated the reversal of the PBG via the reversed barotropic stream function, as suggested by the reviewer. In addition, the heat budget in the cavity has already illustrated the increased heat into the cavity in the 2060s. Therefore, we have decided to remove the analysis of PBG and PBECC heat transport, as it offers less valuable insight compared to the new analysis. The removal of this analysis will not weaken the integrity of this study.

As for the analysis of the heat budget in the cavity, we note the calculations of heat fluxes in the cavity are based on monthly outputs and the issue of $(v \cdot T)^{1mo} \neq v^{1mo} \cdot T^{1mo}$ has not been addressed. However, the object of this study is the AmIS melt rate which exhibits stronger inter-annual to decadal variability rather than variability shorter than one month. The heat budget in the cavity is properly closed (Figure 9), which suggests that $v^{1mo} \cdot T^{1mo}$ may adequately explain the changes in the monthly-mean melt rate. The use of monthly mean outputs will not change our main conclusions. This also quantifies that about 50% of the heat into the cavity is used to melt ice. In addition, Liu et al. (2017) presented a similar analysis of meridional heat transport across a zonal section (73-78°E, 67°S, which is close to the PBG section in our study) with an eddy-resolving model and daily outputs. They decomposed the total meridional heat transport ($Q$) into the mean flow ($Q_m(\overline{v \cdot T}^{1mo})$) and the eddies ($Q - Q_m(\overline{v \cdot T}^{1mo})$) components and suggested that the mean flow components contribute to about 50% of the total meridional heat transport. However, our model resolution is lower, which suggests the contributions from the high-frequency variability may not be as large as those in Liu et al. (2017). Moreover, our focus is the depths below 300 m, which may exhibit smaller variability. It might have a minor impact on the main findings of the study. We could re-run the simulations if the reviewer is still unsatisfied with the monthly data.

– **Show the reversal of PGB with a streamfunction, rather than just two sections of meridional velocity.**
We have shown the reversal of PGB with barotropic stream function (Figure 7; Figure 10d), and we have rewritten Section 3.2.1 presenting the reversal of PBG. Additionally, we have provided time-against-depth plots of V velocity of PBG under two scenarios in Appendix A1.

The causes of PGB reversal are only conjectures in this analysis. For instance, the authors state that PGB is in geostrophic equilibrium due to the salinity difference between two boxes, S1 and S2. However, this is not a rigorous calculation. It merely suggests that it is possible for the gyre to be in geostrophic balance. To make a definitive statement, one would need to consider all the tendency terms of the forces driving the gyre. This approach is not rigorous.

We thank the reviewer for pointing out the inappropriate use of language. We have rephrased the statement of "geostrophic equilibrium/balance" throughout the manuscript, and modified it as:

*"The reversal of PBG is a consequence of the reversal of horizontal salinity/density differences between PBG and its western regions."*

In addition, we have added more analysis of the forcing of PBG (Figure 11). This is to investigate another possible driver for the PBG reversal, for example, surface freshwater forcing from sea ice, Ekman transport, and Ekman pumping, although this does not change our main conclusion that the ice shelf freshwater reverses the horizontal salinity gradients. Closing the vorticity budget of the gyre is out of the research scoop. Following the previous calculation, we presented a scale analysis of Lines 396-438 in the new version of the manuscript with tracked changes.

From lines 290 to 298, the authors claim that the upstream systems influence the ice shelf downstream, but there is no proof provided, only hypotheses.

We will remove the statement about "the influence of upstream systems".

5) Conclusion

– From lines 314 to 323, I am unclear about the authors' intended message. Galton-Fenzi (2009) mentioned that the PBG depends on the relative strength of McKenzie and Barrier polynyas, and it is already known that PBG drives mCDW into the cavity. What new information is being presented here?

We intended to discuss differences of PBG reversal between our study and Galton-Fenzi (2009). However, the discussion is inadequate and not informative. We thank the reviewer for pointing it out. We have added a new analysis of sea ice in the Barrier polynya (Figure 12) and have moved the discussion from Section 4 to Section 3.2.2. As this analysis provides some supporting information for the mechanism of the PBG reversal, this arrangement will make the discussion on the processes more coherent. We summarised the analysis here to save the reviewer's time:

Our model overestimates the strength of the Barrier polynya during the historical period, which suppresses the formation of the clockwise PBG in summer. This is why the seasonality of PBG in our simulations is weaker than in Galton-Fenzi (2009). The reduction in sea ice in the 2040s facilitates the pathway of the clockwise PBG. Although sea ice reduction is not the direct driver of the PBG reversal, it establishes the preconditions for the reversal.

A detailed discussion has been presented from Lines 445-479 in the new version of the manuscript with tracked changes.

– **Lines 324-331 are not informative.**

We have rephrased.

*"A similar mechanism for the regime change is found in Filchner-Ronne Ice Shelf (Hellmer et al., 2012, 2017; Naughten et al., 2021; Siahaan et al., 2022) and Ross Ice Shelf (Siahaan et al., 2022). A redirection of coastal current is driven by a reversed density gradient across the Filchner Trough (Hellmer et al., 2012, 2017; Naughten et al., 2021; Siahaan et al., 2022)/ the little America Basin in the Ross Sea (Siahaan et al., 2022), which facilitates the penetration of mCDW into the ice shelf cavity and causes the regime change. The similar processes found in the sectors for the three large cold ice shelves emphasise the importance of buoyancy changes on the shelf sea in a warming future. This suggests the necessity of long-term records for the shelf sea salinity and sea ice in order to obtain an early warning of the regime change in the three large cold cavities."*

– **Lines 335-349 do not offer conclusions but rather present some reflections, without providing a perspective on future work.**

We have added a perspective on future work

–
*"... A series of freshwater perturbation experiments across various climate scenarios would help address the above questions."*

–
*"... Coupling our standalone ocean configuration with an ice sheet model will be a future step to address the question of the AmIS stability."*

– **Lines 350-380 appropriately discuss the limitations of the work, but the authors overlook the issue of frazil ice refreezing at the ice shelf base, which the model does not account for.**

We have added a discussion about the frazil ice as suggested.

*"... Moreover, we do not include the frazil processes in this configuration. Galton-Fenzi et al. (2012) suggests the inclusion of frazil in the AmIS simulation decreases the melt rates and increases the arrection rates. No frazil ice in our simulation may result in the overestimation."*

**Figures**

The quality of the figures is inadequate. For example, the color bar in Figure 1 does not provide a clear visualization of the bathymetry in the Amery cavity. Additionally, the labels in Figure 2 are so small that they are barely legible, among other issues.

We have updated all figures as suggested.

---

## Author Comment (AC3)

**Supplementary material for 'Current reversal leads to regime change in Amery Ice Shelf cavity in the twenty-first century'**

Jing Jin[1, a], Antony J. Payne[1, a], and Christopher Y. S. Bull[2]

[1]School of Geographical Sciences, University of Bristol, Bristol, UK
[2]Department of Geography and Environmental Sciences, Northumbria University, Newcastle Upon Tyne, UK
[a]now at: Department of Earth, Ocean and Ecological Sciences, University of Liverpool, Liverpool, UK

**S1   Tuning of tidal velocity**

The value of prescribed tidal velocity ($U_t$) in our simulations was determined based on the results from a sensitivity experiment, in which we modelled the melt rate in 2006 by increasing the tidal velocity from 0 cm/s to 12 cm/s in increments of 2.5 cm/s. The melt rate is increased with the increase in tidal velocity (Figure S1a). Figure S1b suggest that the modelled melt rate with tidal velocity of 2.5, 5 and 7.5 cm/s falls within the observational range (Wen et al., 2010; Yu et al., 2010; Depoorter et al., 2013; Rignot et al., 2013; Herraiz-Borreguero et al., 2016; Adusumilli et al., 2020; Rosevear et al., 2022). To a different extent, the three runs overestimate the melt rate north of the grounding line and underestimate the melt rate at the grounding line (not shown). This is the limitation of our configuration. The relatively coarse resolution cannot represent the steep meridional distributions of basal melting. In addition, Rosevear et al. (2022) estimated the magnitude of typical tidal current under Amery Ice Shelf from four tidal constituents. The estimated magnitude is 9.8 cm/s, which yielded an annual-mean Amery melt rate of 1.1 m/yr in 2006. This is slightly higher than the observational estimate. Considering the tidal velocity of 2.5 cm/s is much smaller than the observed tidal velocity, and 7.5 cm/s might largely overestimate the future melt rate given a very high climate sensitivity of UKESM1.0, we decided to use 5 cm/s.

[Figure]

**Figure S1.** (a) Melt rate against tidal velocity ($U_t$). (b) Comparison of the melt rate with different $U_t$ with the observational studies.

**S2 Supplementary figures for the SSP1-2.6 run forced by the r1i1p1f2 ensemble member**

[Figure]

**Figure S2.** Time-mean of temperature, salinity at a transect extending from the grounding line (GL), Prydz Bay Gyre and the shelf break under SSP1-2.6 scenario. The coordinates show the distance from GL against the depth below sea level. The location of the transect is shown by the white line in Figure 1. The years of the time-mean are shown in the title of each panel.

[Figure]

**Figure S3.** Time-mean of temperature against salinity diagram for each grid cell within the ice shelf cavity (the solid dots) and on the continental shelf (the open triangles) under the SSP1-2.6 scenario. The grey solid lines are potential density. The black dashed line shows the surface freezing point. The black solid line represents the Gade line, defined by end members of the temperature and salinity of the warmest water mass (the black pentacle). The colour schemes show the depth of each grid cell. The years of the time-mean are shown in the title of each panel.

[Figure]

**Figure S4.** Timeseries of depth-mean (a) temperature and (b) salinity averaged within the ice shelf cavity (the green line) and on the continental shelf below 300 m (the orange line) under the SSP1-2.6 scenario. The vertical dashed line indicates the year of 2015. A 12 month running average is applied.

[Figure]

**Figure S5.** Time-mean of barotropic streamfunction (BSF, in units of Sv) under the SSP1-2.6 scenario. The red-blue colour schemes are for BSF on the open ocean. The values of BSF contours on the open ocean are -5, -4, -3, -2, -1.5, -1, 0, 1, 1.5, 2 Sv. The orange-purple colour schemes are for BSF in the ice shelf cavities. The values of BSF contours in the cavities -0.9, -0.6, -0.3, -0.1, 0, 0.1, 0.2 Sv. The positive/negative BSF represent anti-clockwise/clockwise circulation. The solid and dotted purple line shows the PBG transect and the extended transects to the coasts. The years of the time-mean are shown in the title of each panel.

[Figure]

**Figure S6.** Timeseries of (a) integrated ice shelf freshwater fluxes ($10^4$ m·s$^{-3}$), (b) integrated volume of sea ice on the continental shelf ($10^{11}$ m$^3$), (c) area-mean zonal surface stress within the PBG area ($10^{-2}$ N·m$^{-2}$), (d) area-mean meridional surface stress within the PBG area ($10^{-2}$ N·m$^{-2}$), (e) area-mean Ekman pumping within the PBG area ($10^{-6}$ m·s$^{-1}$) under the SSP1-2.6 scenario. The PBG area is defined using the closed BSF contour of -1.5 Sv in front of the ice shelf in Figure 7f. The vertical dashed line indicates the year of 2015. A 12 month running average is applied.

**S3 Supplementary figures for the r2i1p1f2 ensemble member**

[Figure]

**Figure S7.** Timeseries of area-mean of mid-depth temperature and salinity within Prydz Bay across five UKESM1.0-LL ensemble members under the SSP5-8.5 and SSP1-2.6 scenario. Note the temperature difference between the r1 and r2 (the thicker lines).

[Figure]

**Figure S8.** The r2-AME025 results. (a) Time series of the area-averaged melt rate $(\mathrm{m \cdot yr^{-1}})$ from 1976 to 2100. A 12-month-running-average is applied. The dashed vertical line indicates the start of 2015. The time-averaged basal melting over the period of (b) 1976-2014 (Historical), (c) 2076-2100 under SSP5-8.5, (d) 2076-2100 under SSP1-2.6. The warm/cold colours represent melting/refreezing, respectively. Note the different colormap ranges. The colormap for (c) is saturated.

To exclude the abrupt increase in melt rate and the regime change not due to the instability of the model itself, we conducted a second series of experiments forced by the r2i1p1f2 ensemble member of UKESM1.0-LL. Figure S7 shows the comparisons of area-mean mid-depth temperature and salinity within Prydz Bay across five ensembles under the SSP5-8.5 and SSP1-2.6 scenarios. Temperature and salinity changes under the SSP5-8.5 scenario across the ensembles are similar (Figure S7a, b). What is striking is the temperature under the SSP1-2.6 scenario for the r2 ensemble (The thick red line, Figure S7c). The increase in temperature for the r2 ensemble starts in the 2050s-2060s, approximately 20 years later than the temperature increase in the r1 ensemble, which starts in the 2030s (Figure S7c)). Therefore, we chose the r2 ensemble as the forcing of the second series of the AME025 simulations (hereafter the r2-AME025 simulations).

Figure S8 shows that the AmIS melt rate under the SSP1-2.6 scenario from the r2-AME025 simulation behaves similarly to that from the r1-AME025 simulation (Figure 3). However, the timing of the increase occurs later, which coincides with the temperature differences between the r1 and r2 UKESM1.0-LL ensembles. The time evolution of the shelf sea temperature and the cavity temperature under the SSP1-2.6 scenario (Figure S9), and the time evolution of the properties at the PBG transect under the SSP1-2.6 scenario (Figure S9) behave in the same way as the AmIS melt rate. This suggests that the variability of AmIS melt rate and PBG is ultimately controlled by the oceanic forcing, especially the temperature forcing.

In addition, the abrupt increase in the melt rate, the delayed warming in the cavity, the reversed salinity/density gradients between PBG and its western regions and the reversal of PBG are seen in the r2-AME025 simulations under both scenarios (Figure S8-S12). This supports our main conclusions that the freshening-driven PBG reversal causes the regime change in the AmIS cavity.

[Figure]

**Figure S9.** The r2-AME025 results. The outputs are taken from the SSP1-2.6 run. Timeseries of depth-mean (a) temperature and (b) salinity averaged within the ice shelf cavity (the green line) and on the continental shelf below 300 m (the orange line). The vertical dashed line indicates the year of 2015. A 12 month running average is applied.

[Figure]

**Figure S10.** The r2-AME025 results. Hovemoller diagram of properties at the zonal transect shown in Figure 7a. The outputs are taken from the SSP1-2.6 run. (a) Depth-mean temperature below 300 m. (b) Depth-mean salinity below 300 m. The potential density below 300 m overlies temperature and salinity. The values of potential density contours are 27.3, 27.4, 27.5, 27.6, 27.7, 27.8 kg·m$^{-3}$. (c) Zonal sea surface height gradients (SSH gradient). A positive/negative SSH gradient indicates that the SSH in the east is higher/lower than in the west. The values of SSH gradient contours are -3, -2.5, -1.5, -0.5, 0, 0.25, 1, 2 mm·km$^{-1}$. (d) Barotropic streamfunction (BSF). The values of BSF contours are -2.5, -2, -1.5, -1, -0.5, 0, 0.25, 0.5, 1 Sv. A positive/negative BSF represents anti-clockwise/clockwise circulation. The transect in between the purple dashed lines is the PBG transect shown by the solid purple line in Figure 7a. A 12 month running average is applied.

[Figure]

**Figure S11.** The r2-AME025 results. The outputs are taken from the SSP5-8.5 run. Timeseries of depth-mean (a) temperature and (b) salinity averaged within the ice shelf cavity (the green line) and on the continental shelf below 300 m (the orange line). The vertical dashed line indicates the year of 2015. A 12 month running average is applied.

[Figure]

**Figure S12.** The r2-AME025 results. Hovemoller diagram of properties at the zonal transect shown in Figure 7a. The outputs are taken from the SSP5-8.5 run. (a) Depth-mean temperature below 300 m. (b) Depth-mean salinity below 300 m. The potential density below 300 m overlies temperature and salinity. The values of potential density contours are 27.3, 27.4, 27.5, 27.6, 27.7, 27.8 kg·m$^{-3}$. (c) Zonal sea surface height gradients (SSH gradient). A positive/negative SSH gradient indicates that the SSH in the east is higher/lower than in the west. The values of SSH gradient contours are -3, -2.5, -1.5, -0.5, 0, 0.25, 1, 2 mm·km$^{-1}$. (d) Barotropic streamfunction (BSF). The values of BSF contours are -2.5, -2, -1.5, -1, -0.5, 0, 0.25, 0.5, 1 Sv. A positive/negative BSF represents anti-clockwise/clockwise circulation. The transect in between the purple dashed lines is the PBG transect shown by the solid purple line in Figure 7a. A 12 month running average is applied.

**References**

Adusumilli, S., Fricker, H. A., Medley, B., Padman, L., and Siegfried, M. R.: Interannual variations in meltwater input to the Southern Ocean from Antarctic ice shelves, NATURE GEOSCIENCE, 13, 616+, https://doi.org/10.1038/s41561-020-0616-z, 2020.

Depoorter, M. A., Bamber, J. L., Griggs, J. A., Lenaerts, J. T. M., Ligtenberg, S. R. M., van den Broeke, M. R., and Moholdt, G.: Calving fluxes and basal melt rates of Antarctic ice shelves, NATURE, 502, 89+, https://doi.org/10.1038/nature12567, 2013.

Herraiz-Borreguero, L., Church, J. A., Allison, I., Peña-Molino, B., Coleman, R., Tomczak, M., and Craven, M.: Basal melt, seasonal water mass transformation, ocean current variability, and deep convection processes along the Amery Ice Shelf calving front, East Antarctica, Journal of Geophysical Research: Oceans, 121, 4946–4965, https://doi.org/https://doi.org/10.1002/2016JC011858, 2016.

Rignot, E., Jacobs, S., Mouginot, J., and Scheuchl, B.: Ice-Shelf Melting Around Antarctica, SCIENCE, 341, 266–270, https://doi.org/10.1126/science.1235798, 2013.

Rosevear, M., Galton-Fenzi, B., and Stevens, C.: Evaluation of basal melting parameterisations using in situ ocean and melting observations from the Amery Ice Shelf, East Antarctica, Ocean Science, 18, 1109–1130, https://doi.org/10.5194/os-18-1109-2022, 2022.

Wen, J., Wang, Y., Wang, W., Jezek, K., Liu, H., and Allison, I.: Basal melting and freezing under the Amery Ice Shelf, East Antarctica, Journal of Glaciology, 56, 81–90, https://doi.org/10.3189/002214310791190820, 2010.

Yu, J., Liu, H., Jezek, K. C., Warner, R. C., and Wen, J.: Analysis of velocity field, mass balance, and basal melt of the Lambert Glacier–Amery Ice Shelf system by incorporating Radarsat SAR interferometry and ICESat laser altimetry measurements, Journal of Geophysical Research: Solid Earth, 115, https://doi.org/https://doi.org/10.1029/2010JB007456, 2010.

---

## Editor Decision (ED1)

**Editor's final comments on EGUSPHERE-2024-1287**

\*\*Line numbers refer to the track-change version\*\*

L. 13-14: reformulate or explain "which makes it susceptible to ocean temperature changes in the deep ice shelf cavity, due to a low in-situ freezing point temperature". Do you mean that because of the quadratic dependency to the thermal forcing, a low freezing point implies that any ocean temperature change has a stronger effect on melt rates than with a higher freezing temperature? (with a linear dependency, this would not be the case).

I find it hard to read the text with all the acronyms. While it is all right to indicate PB for Prydz Bay in a small figure, I think the text reads better if you use "Prydz Bay" in the whole text. Same for PC, AD, FB, FLB, GL. Ok for PBG that is used a lot and for PBECC that is quite long.

L. 23-24: I am not a native speaker but "HSSW is dense and cold (slightly below the surface freezing point), which forms in coastal polynyas within PB during sea ice formation" would probably be more correct as something like "HSSW is a dense and cold (slightly below the surface freezing point) water mass that forms in coastal polynyas within Prydz Bay during sea ice formation".

L. 38: "a freshening in PB increases vertical stratification, and induces the warming" would be clearer as something like "a freshening at the surface of Prydz Bay increases vertical stratification and induces warming at depth".

L. 68: Gurvan et al. (2017) should be Madec et al. (2017), Gurvan is his first name.

L. 93: "The top boundary is set at 30 m" -> The top boundary-layer thickness is set to 30 m.

L. 99-100: Are you sure that there is salt restoring in the ocean of UKESM1.0-LL? Usually, salt restoring is only applied in the standalone ocean configurations, not in the coupled models. Not sure that GO7 is relevant here if it is the standalone configuration (and if it is, GO6 rather than GO7 is mentioned in section 2.2). Please clarify this point. See also comment about L. 138.

L. 102: "as they are prescribed in the UKESM1.0-LL outputs" -> "as they are in the UKESM1.0-LL outputs.

L. 98-102: this paragraph could be moved to section 2.3 where further information on the boundary conditions is provided.

Section 2.2: Consider pointing to Caillet et al. (2025) in which UKESM is very well ranked (Fig. 1 of their preprint) and in which an important natural variability is emphasized in front of Amery (Fig. 3 of their preprint). https://doi.org/10.5194/egusphere-2024-128

L. 136: "optimistic" is somewhat subjective -> low emission

L. 138: is GO7 a standalone ocean simulation? Please provide information on why you did not take the ocean outputs of UKESM1.0-LL?

L. 148: the model drift for which variable?

L. 190-191: "This suggests that our regional model […] has a stronger response to climate warming" -> This suggests that our regional model […] produces a stronger climate warming (or a stronger response to increasing emissions of greenhouse gases).

L. 166-170: it would be more robust to indicate mean values and intervals of confidence over a given period than approximated values at single dates. Same for the conclusion L. 5115-520.

L. 221: "the SSP5-8.5" -> "the SSP5-8.5 scenario" or "SSP5-8.5".

L. 221: the definition of water massES.

L. 260 "the two salinity" -> the two salinities (or the two salinity values)

L. 261: divergent -> diverge.

L. 262: "The difference is likely due to the reduction in sea ice". First, you should indicate "reduction in sea ice production" which is what matters for dense water formation. Second, this may not be the only reason: reduced vertical convection could also come from a more stratified ocean due to more freshwater released at the surface of the Southern Ocean in UKESM (for the wrong reason that the ice sheet mass is kept constant, i.e. that additional snowfall on a warmer ice sheet is injected into the ocean). The lateral boundary could propagate this signal in the regional domain.

L. 296: transition -> transitions?

L. 297: "HSSW becoming less efficient in driving the melting": what does this mean physically?

L. 326: the units should be J K$^{-1}$ kg$^{-1}$.

L. 328: provide units for "334".

L. 342-343: I don't understand "its main flow travels offshore" here. What does this mean for a gyre?

Fig. 11b: The net sea ice production over the continental shelf would have been more informative than the sea ice volume, and more similar to the quantity used for ice shelves in Fig. 11a.

Fig. 11c-e and related text: clarify whether Ekman pumping and the surface stresses are calculated from the stress at the ocean surface, i.e., either from wind or from sea-ice.

Eq. (6): shouldn't there be a minus sign on the last term? Why not directly write the equation without the neglected effect of wind stress? If formulated like this, I think that $\tau_x$ should be called the Reynolds stress ($\rho\langle u'w'\rangle$) rather than the surface stress.

Eq. (7): shouldn't there be an SSH term ($+\rho_{ref}g\ dSSH$)? Could this be used to discuss the expected wind effect compared to the thermohaline effect?

L. 452-453: "overestimated sea ice" what? Production, concentration, …? Same with the overestimation L. 454-455, with the reduction L. 471, with the decrease L. 476-477.

L. 478: "are more possible" -> exist

L. 502: "the r2 ensemble of UKESM" -> The r2 member of the UKESM1.0-LL ensemble.

L. 502-503: "the r2 ensemble" -> the r2 member ; "the r1 ensemble" -> the r1 member.

L. 580-584: Rosevear et al. (2022) reported a 400% overestimation of the three equations of Jenkins et al. (2010). Here, in NEMO, the value of $\sqrt{C_d}\Gamma_T$ is 1.6 times smaller than in Jenkins et al. (2010). But Rosevear et al. (2022) also used a different velocity in the three equations (averaged from 7m to 19m below the ice) than what is implemented in NEMO. In addition, you tuned the tidal velocity to match observational melt rates (Fig. S1). Therefore, while it is important to raise the caveat of the melt calculation and to cite Rosevear et al. (2022), I would not mention the value of the overestimation (200-400%).

---

## Author Response (AR2)

**Responses to the editor**

We appreciate the detailed comments from the editor Nicolas Jourdain. Below we copy the editor's comments in **bold black** and write our responses in **bold blue**. The changes to the manuscript are in *italic blue*.

**L. 13-14: reformulate or explain "which makes it susceptible to ocean temperature changes in the deep ice shelf cavity, due to a low in-situ freezing point temperature". Do you mean that because of the quadratic dependency to the thermal forcing, a low freezing point implies that any ocean temperature change has a stronger effect on melt rates than with a higher freezing temperature? (with a linear dependency, this would not be the case).**

**Thanks for the suggestion. We have reformulated this sentence.**

*"This makes AmIS susceptible to ocean temperature changes in the deep ice shelf cavity (Galton-Fenzi et al., 2012; Wang et al., 2022). As ice shelf melt rates quadratically respond to external thermal forcing (Holland et al., 2008), ice with a low freezing point implies that it is more sensitive to any ocean temperature change than with a higher freezing point."*

**I find it hard to read the text with all the acronyms. While it is all right to indicate PB for Prydz Bay in a small figure, I think the text reads better if you use "Prydz Bay" in the whole text. Same for PC, AD, FB, FLB, GL. Ok for PBG that is used a lot and for PBECC that is quite long.**

**Thanks for the suggestion. We have removed all the acronyms for the geographic features in the text expect PBG.**

**L. 23-24: I am not a native speaker but "HSSW is dense and cold (slightly below the surface freezing point), which forms in coastal polynyas within PB during sea ice formation" would probably be more correct as something like "HSSW is a dense and cold (slightly below the surface freezing point) water mass that forms in coastal polynyas within Prydz Bay during sea ice formation".**

**Thanks. We have corrected the sentence.**

**L. 38: "a freshening in PB increases vertical stratification, and induces the warming" would be clearer as something like "a freshening at the surface of Prydz Bay increases vertical stratification and induces warming at depth".**

**Thanks. We have rephrased the sentence.**

**L. 68: Gurvan et al. (2017) should be Madec et al. (2017), Gurvan is his first name.**

**Done.**

**L. 93: "The top boundary is set at 30 m" -> The top boundary-layer thickness is set to 30 m.**

**Done.**

**L. 99-100: Are you sure that there is salt restoring in the ocean of UKESM1.0-LL? Usually, salt restoring is only applied in the standalone ocean configurations, not in the coupled models. Not sure that GO7 is relevant here if it is the standalone configuration (and if it is, GO6 rather than GO7 is mentioned in section 2.2). Please clarify this point. See also comment about L. 138.**

Thanks for pointing out the mistake. No, there is no salt restoring in UKESM1.0-LL. The SSS restoration is applied in GO7. However, it is irrelevant to this work. We have removed the sentence "However, the GO7 configuration—the ocean core of UKESM1.0-LL—has restoring in the global domain (Storkey et al., 2018)."

GO7 is a standalone configuration. It is identical to the GO6 0.25° configuration except ice shelf cavities are open in GO7. We have added descriptions about the difference between GO6 and GO7 in the text:

*"The difference between GO6 and GO7 mentioned before is that GO7 is identical to a higher resolution GO6 0.25° configuration except there are open ice shelf cavities in GO7 (Storkey et al., 2018). The differences between the 1° and 0.25° GO6 configurations are the mixing and boundary conditions, which are adjusted according to time step and grid spacing (Storkey et al., 2018)."*

**L. 102: "as they are prescribed in the UKESM1.0-LL outputs" -> "as they are in the UKESM1.0-LL outputs.**

Done.

**L. 98-102: this paragraph could be moved to section 2.3 where further information on the boundary conditions is provided.**

Done.

**Section 2.2: Consider pointing to Caillet et al. (2025) in which UKESM is very well ranked (Fig. 1 of their preprint) and in which an important natural variability is emphasized in front of Amery (Fig. 3 of their preprint). https://doi.org/10.5194/egusphere-2024-128**

We have cited Caillet et al. (2024) in Section 2.2:

*"UKESM1.0-LL is well ranked in terms of various Southern Ocean and Antarctic sea properties (Caillet et al., 2024, Figure 1)."*

*"UKESM1.0-LL also exhibits a large internal climate variability of salinity averaged between 200 m and 700 m in front of AmIS, which is important for HSSW formation in Prydz Bay (Caillet et al., 2024, Figure 3c)."*

**L. 136: "optimistic" is somewhat subjective -> low emission**

Done.

**L. 138: is GO7 a standalone ocean simulation? Please provide information on why you did not take the ocean outputs of UKESM1.0-LL?**

**We do not use UKESM1.0-LL mainly because there are no open ice shelf cavities in it. We have provided the information:**

*"The reason we chose GO7 rather than UKESM1.0-LL is that open ice shelf cavities are presented in G07 but not in UKESM1.0-LL. The oceanographic properties inside the AmIS cavity and in the open ocean are more physically consistent if the initial conditions are taken from one dataset."*

**L. 148: the model drift for which variable?**

**We have added the information:** *"...until the model drift for ice shelf melt rate, ocean temperature and salinity becomes small (Figure 2a, b)."*

**L. 190-191: "This suggests that our regional model [. . . ] has a stronger response to climate warming" -> This suggests that our regional model [. . . ] produces a stronger climate warming (or a stronger response to increasing emissions of greenhouse gases).**

**Done.**

**L. 166-170: it would be more robust to indicate mean values and intervals of confidence over a given period than approximated values at single dates. Same for the conclusion L. 5115-520.**

**Thanks for the suggestion. We have rephrased this paragraph and provided the projecting numbers here as well as in the conclusion as suggested.**

*"...The modelled melt rate is $0.75 \pm 0.15 \ \mathrm{m \cdot yr^{-1}}$ (with 99.9% confidence intervals) over the period of 1976-2014. It is drastically increased to $13.14 \pm 2.36 \ \mathrm{m \cdot yr^{-1}}$ over the period of 2076-2100 under the SSP5-8.5 scenario or $8.10 \pm 1.53 \ \mathrm{m \cdot yr^{-1}}$ over the same period under the SSP1-2.6 scenario."*

**L. 221: "the SSP5-8.5" -> "the SSP5-8.5 scenario" or "SSP5-8.5".**

**Done.**

**L. 221: the definition of water massES.**

**Done.**

**L. 260 "the two salinity" -> the two salinities (or the two salinity values)**

**Done.**

**L. 261: divergent -> diverge.**

**Done.**

**L. 262:** "The difference is likely due to the reduction in sea ice". First, you should indicate "reduction in sea ice production" which is what matters for dense water formation. Second, this may not be the only reason: reduced vertical convection could also come from a more stratified ocean due to more freshwater released at the surface of the Southern Ocean in UKESM (for the wrong reason that the ice sheet mass is kept constant, i.e. that additional snowfall on a warmer ice sheet is injected into the ocean). The lateral boundary could propagate this signal in the regional domain.

Thanks for providing another explanation. We have rephrased as suggested.

*"The difference is likely due to the reduction in sea ice production, which matters in dense water formation, while the increased mCDW is present on the continental shelf but has not yet reached the deeper cavity (Figure 5b, c). Reduced vertical convection resulting from a more stratified ocean may also cause the difference. Freshwater forcing from ice sheets is prescribed assuming the ice sheet mass is kept constant in UKESM1.0-LL (Sellar et al., 2020), which results in an additional snowfall on warmer ice sheets and increased freshwater released at the ocean surface. This bias is propagated through the lateral boundary to the regional domain."*

**L. 296:** transition -> transitions?

Done.

**L. 297:** "HSSW becoming less efficient in driving the melting": what does this mean physically?

It is supposed to be "HSSW becoming less efficient in driving the circulation". We have rephrased it:

*"This is accompanied by ..., likely due to reduced formation of HSSW outside of the AmIS cavity (Figure 5b, c), which becomes less efficient at driving the cavity circulation."*

**L. 326:** the units should be J K-1 kg-1

Done.

**L. 328:** provide units for "334".

Done.

**L. 342-343:** I don't understand "its main flow travels offshore" here. What does this mean for a gyre?

We have rephrased this sentence.

*"PBG is anticlockwise and weak before the 2050s, with off-shore transport at PBG transect (Figure 7a-c)."*

**Fig. 11b:** The net sea ice production over the continental shelf would have been more informative than the sea ice volume, and more similar to the quantity used for ice shelves in Fig. 11a.

Thanks for the suggestion. We have replaced the sea ice volume with the net sea ice production in Figure 11b and Figure S6b.

**Fig. 11c-e and related text: clarify whether Ekman pumping and the surface stresses are calculated from the stress at the ocean surface, i.e., either from wind or from sea-ice.**

Thanks. We have clarified that the surface stresses refer to the stress at the ocean surface in the text.

**Eq. (6): shouldn't there be a minus sign on the last term? Why not directly write the equation without the neglected effect of wind stress? If formulated like this, I think that $\tau_x$ should be called the Reynolds stress ($\rho\langle u'w'\rangle$) rather than the surface stress.**

Thanks for pointing out the error. We have removed the stress term as suggested.

**Eq. (7): shouldn't there be an SSH term ($+\rho_{ref}gdSSH$)? Could this be used to discuss the expected wind effect compared to the thermohaline effect?**

Thanks for the suggestion. We have reformulated Eq. 7 as suggested to discuss the effect of SSH and salinity changes. We do not think the SSH term is solely controlled by winds, as the pattern of SSH gradient (Figure 10c) does not match well with the surface stresses (Figure 11c, d). Although we noted the surface stresses is different with the wind stresses, this may not fully represent the effect of surface wind. The changes in SSH gradient (Figure 10c) exhibit consistence with the changes in salinity (Figure 10b, mainly due to the outflow of the ice shelf meltwater), sea ice production (Figure 11b) and Ekman pumping (Figure 11e). It may suggest that the combined effect of the meltwater outflow, sea ice and surface winds causes the SSH change. To align with the modification of Eq. 7, we have also changed Eq.9 and Figure A1, and rephrased the paragraph to discuss the effect of salinity gradient and SSH gradient on the gyre reversal:

*"The estimates are shown in Figure A1. The term of salinity gradient ($\frac{Hb}{\rho_{ref}}\frac{dS}{dx}$) has a magnitude of $8\times10^{-6}$ (Figure A1a). The magnitude of the horizontal SSH gradient ($\frac{dSSH}{dx}$) is up to $4\times10^{-6}$ (Figure A1b), which is less than 50% of the salinity effect. The reconstructed meridional velocity based on Eq 9 agrees with the modelled velocity (Figure A1c). This analysis suggests that the effect of salinity gradient and SSH gradient compete (Figure A1a, b). However, the reversal of PBG is a consequence of the reversal of horizontal salinity differences between PBG and its western regions."*

**L. 452-453: "overestimated sea ice" what? Production, concentration, ...? Same with the overestimation L. 454-455, with the reduction L. 471, with the decrease L. 476-477.**

We have clarified the sea ice variable in the text.

**L. 478: "are more possible" -> exist**

Done.

**L. 502: "the r2 ensemble of UKESM" -> The r2 member of the UKESM1.0-LL ensemble.**

Done.

**L. 502-503: "the r2 ensemble" -> the r2 member ; "the r1 ensemble" -> the r1 member.**

**Done.**

**L. 580-584: Rosevear et al. (2022) reported a 400% overestimation of the three equations of Jenkins et al. (2010). Here, in NEMO, the value of $\sqrt{C_d}\Gamma_T$ is 1.6 times smaller than in Jenkins et al. (2010). But Rosevear et al. (2022) also used a different velocity in the three equations (averaged from 7m to 19m below the ice) than what is implemented in NEMO. In addition, you tuned the tidal velocity to match observational melt rates (Fig. S1). Therefore, while it is important to raise the caveat of the melt calculation and to cite Rosevear et al. (2022), I would not mention the value of the overestimation (200-400%).**

Thanks for the suggestion. We have removed the value.